# Semantic Cache Distillation:
# Efficient State Transfer via Reuse and Selective Patching

Qianli Ma [1 2]  Zhiqing Tang[✉ 2]  Hanshuai Cui [3 2]  Zhi Yao [3 2]  Weijia Jia[✉ 2 4]

## Abstract

Disaggregated serving alleviates memory bottlenecks in Large Language Model (LLM) inference but creates a severe communication bottleneck: transmitting high-dimensional Key-Value (KV) caches often dominates time-to-first-token (TTFT). Moreover, reusing caches across heterogeneous models (e.g., base and fine-tuned variants) causes semantic misalignment that accumulates over layers, degrading generation quality. We propose Semantic Cache Distillation (SCD), a loss-constrained framework that replaces raw KV transmission with compact semantic codes. SCD addresses these challenges via two mechanisms: (1) REUSE, which reconstructs most layers from low-rank subspaces to minimize transfer cost, and (2) PATCH, which predicts normalized inputs at sparse transition layers to truncate error propagation. Empirically, SCD delivers up to 2.65× TTFT speedup over the oracle consumer prefill and dominates quantization and selective recomputation baselines on the quality–latency Pareto frontier in bandwidth-constrained regimes, while keeping generation quality within 5% F1 of the oracle.

## 1. Introduction

LLM serving is increasingly constrained by memory bandwidth and communication overheads rather than compute intensity. In decoder-only Transformers, autoregressive decoding reuses per-layer KV caches to avoid recomputing attention; however, these caches grow linearly with context length and depth. Modern systems therefore disaggregate inference into a compute-bound prefill stage and a memory-bound decode stage, placing them on separate devices (Qin et al., 2024; Zhong et al., 2024; Patel et al., 2024). While advanced schedulers (Agrawal et al., 2024) and memory managers (Kwon et al., 2023) address local efficiency, disaggregation introduces a critical network bottleneck: data transfer between the prefill producer and the decode consumer can dominate TTFT, especially over slow interconnects.

Prior work primarily focuses on compressing the KV cache footprint. Techniques include quantization to compress states into low precision (Liu et al., 2024c; Hooper et al., 2024), sparsification to evict less salient tokens (Zhang et al., 2023; Li et al., 2024a; Tang et al., 2024; Cai et al., 2024b), and low-rank methods that exploit redundancy in Transformer representations. Systems like CacheGen (Liu et al., 2024b) further optimize the streaming of these compressed states. However, these existing approaches typically assume the producer and consumer share the same latent space (i.e., identical weights). They rely on static compression policies that ignore the representation discrepancies inherent in cross-model adaptation.

SCD targets the narrower but practical case of producer-consumer pairs that share the same Transformer architecture while differing in weights. It is not intended to replace same-model KV reuse, where raw or quantized caches can already be reused directly. Instead, SCD addresses shared-architecture, weight-mismatched serving, such as a generic base model serving as the producer for fine-tuned specialists (Sheng et al., 2024; Chen et al., 2024) or a draft/verifier pair (Cai et al., 2024a; Li et al., 2024b).

A representative deployment is *shared-prefill, specialized-decode* serving. For example, an online service may run a shared producer on a long common prefix, such as user history, documents, instructions, or a task context, and then route the resulting state to task-specialized consumers for tutoring, code assistance, safety filtering, or domain-specific response generation. The producer and consumers are intentionally not identical: specialization is the purpose of the

---

[1]Faculty of Arts and Science, Beijing Normal University, Zhuhai 519087, China [2]Institute of Artificial Intelligence and Future Networks, Beijing Normal University, Zhuhai 519087, Guangdong, China [3]School of Artificial Intelligence, Beijing Normal University, Beijing 100875, China [4]Guangdong Key Lab of AI and Multi-Modal Data Processing, Beijing Normal-Hong Kong Baptist University, Zhuhai 519087, China. Correspondence to: Zhiqing Tang <zhiqingtang@bnu.edu.cn>, Weijia Jia <jiawj@bnu.edu.cn>.

*Proceedings of the 43rd International Conference on Machine Learning*, Seoul, South Korea. PMLR 306, 2026. Copyright 2026 by the author(s).

back-end models. Once their weights differ, however, the producer's raw KV states are no longer natively compatible with the consumer. This setting presents two coupled challenges: (1) **Transmission Overhead:** Raw KV transfer is bandwidth-bound in disaggregated serving, so any practical system must shrink the per-request payload while preserving the consumer's distribution. (2) **Semantic Drift:** Weight discrepancies cause representation mismatches that compound through layers; direct reuse degrades quality, while full recomputation negates the latency benefits of disaggregation. Existing approaches (Liu et al., 2024a) attempt to mitigate this by selectively recomputing layers or sharing only specific pairs. However, these methods enforce a rigid trade-off: prioritizing reuse risks quality, while prioritizing recomputation sacrifices latency. A practical solution must simultaneously minimize transmission costs and correct drift under a fixed online execution path.

This deployment pattern is attractive precisely because the expensive prefix computation can be amortized across a family of specialized consumers. Synchronizing all endpoints to identical weights would remove the specialization that motivates the system, while colocating every specialized model with the producer would duplicate memory and undermine disaggregation. The relevant question is therefore not whether same-model KV reuse can work, but how to transfer state when architectural compatibility exists while the consumer's weight space is different.

We propose **Semantic Cache Distillation (SCD)**, a bandwidth-efficient state-transfer framework that replaces raw KV transmission with compact semantic codes for shared-architecture, weight-mismatched producer-consumer pairs. SCD distills high-dimensional states at the producer and reconstructs consumer-aligned states via lightweight, layer-aware translators. SCD integrates two complementary mechanisms: (1) REUSE: Leveraging the high cross-model compatibility of most layers, SCD reconstructs states via fast low-rank projections to minimize bandwidth usage. (2) PATCH: For the few critical layers that induce significant drift, SCD predicts normalized pre-attention inputs to truncate error propagation without full recomputation. In summary, we make the following contributions:

- **SCD Framework.** We propose an end-to-end design that integrates REUSE for bandwidth efficiency and PATCH for semantic alignment, enabling low-latency transfer between differing models.

- **Heterogeneous State Transfer Formulation.** We formalize the problem of cache REUSE under representation mismatch, identifying why raw transfer and static compression fail in disaggregated settings.

- **Selective Correction Mechanism.** We introduce PATCH, a lightweight module applied at sparse transi-

tion layers that truncates error propagation with minimal computational overhead.

- **Empirical Effectiveness.** We demonstrate that SCD achieves superior latency-quality trade-offs in bandwidth-limited environments, delivering up to $2.65\times$ TTFT speedup over the oracle consumer prefill while maintaining generation quality and outperforming quantization and DroidSpeak-style recomputation baselines on the Pareto frontier.

**Conflict of Interest Disclosure.** The authors declare no financial conflicts of interest related to this work.

## 2. Related Work

Our work intersects with disaggregated LLM serving, efficient KV cache management, and cross-model alignment.

**Disaggregated Serving and the Transfer Bottleneck.** Modern serving systems separate *prefill* and *decode* phases across distinct workers to maximize utilization (Zhong et al., 2024; Qin et al., 2024; Patel et al., 2024). While this architecture improves throughput, it places KV-cache transfer on the critical path of time-to-first-token (TTFT). Advanced schedulers like Sarathi-Serve (Agrawal et al., 2024) and FastServe (Wu et al., 2023) use chunk-level scheduling to mitigate stalls, while engines like vLLM (Kwon et al., 2023) and SGLang (Zheng et al., 2024) optimize memory fragmentation. Recent transport-layer works, such as FlowKV (Li et al., 2025), optimize low-latency KV-cache transfer and load-aware scheduling. However, these system-level optimizations largely treat transferred states as *system-level payloads* rather than learning cross-model semantic translators. Our work complements these efforts by optimizing the *payload*: we target the producer-to-consumer handoff, particularly when raw transfer between heterogeneous models becomes bandwidth-prohibitive.

**Intra-Model KV Compression.** Standard techniques to reduce cache footprint include quantization methods like KIVI (Liu et al., 2024c) and KVQuant (Hooper et al., 2024). Beyond quantization, pruning strategies such as H2O (Zhang et al., 2023), SnapKV (Li et al., 2024a), Quest (Tang et al., 2024), and PyramidKV (Cai et al., 2024b) evict less salient tokens, while StreamingLLM (Xiao et al., 2023) uses attention sinks for infinite-length inference. Systems like CacheGen (Liu et al., 2024b) further optimize the streaming transmission of compressed states. Crucially, these approaches are inherently *intra-model*: they assume the producer and consumer share identical weights and feature spaces. Applying them directly to heterogeneous pairs fails to bridge the semantic gap caused by weight discrepancies. In contrast, SCD is designed for the *inter-model*

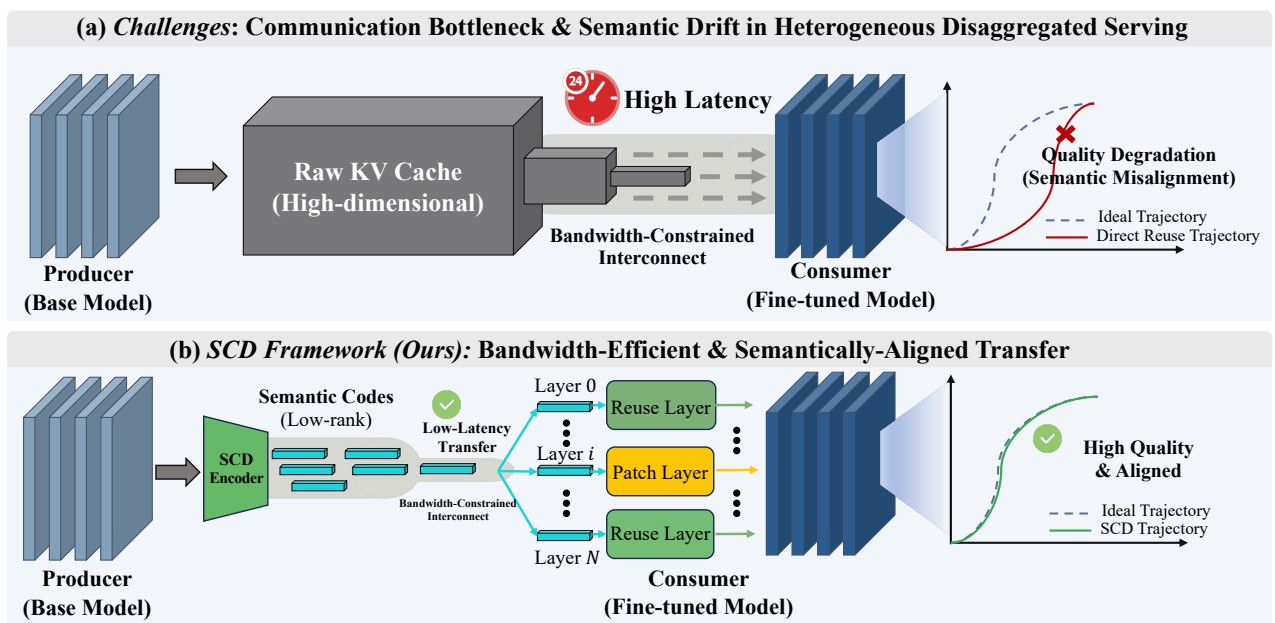

*Figure 1.* Overview of SCD. (a) Challenges in heterogeneous disaggregated serving: Transmitting raw KV caches creates a communication bottleneck, and directly reusing caches from a base model (Producer) to a fine-tuned model (Consumer) causes semantic drift that degrades quality. (b) SCD Framework: We replace raw KV transmission with compact semantic codes. The Consumer reconstructs states using REUSE (low-rank projection) for efficiency and applies PATCH at sparse transition layers to rectify semantic misalignment, achieving low latency and high generation quality.

setting, mapping producer states into the consumer's native space via learnable codes.

**Cross-Model Cache Reuse.** Serving heterogeneous models with a shared backbone is a growing challenge. Systems like S-LoRA (Sheng et al., 2024) and Punica (Chen et al., 2024) enable scalable serving of LoRA adapters, while Prompt Cache (Gim et al., 2024) explores attention reuse across requests. For reuse between base and fine-tuned models, Liu et al. (2024a) show that naive sharing degrades generation quality. Their system, DroidSpeak, mitigates this by selectively recomputing critical layers while reusing raw caches for the rest. Unlike DroidSpeak, which makes a binary decision (reuse raw vs. recompute), SCD advances from *selection* to *transformation*. By employing loss-constrained reconstruction (Reuse) and targeted correction (Patch), SCD achieves higher compression than raw reuse and lower latency than recomputation, smoothing the trade-off frontier.

**Orthogonal Acceleration Approaches.** SCD focuses on state transfer and is orthogonal to decoding-stage optimizations. Techniques like CoDec (Wang et al., 2025b) accelerate prefix-shared decoding kernels, while speculative decoding frameworks like Medusa (Cai et al., 2024a) and EAGLE (Li et al., 2024b) generate multiple tokens per step using draft models. SCD is compatible with these methods: once the cache is transferred and reconstructed, the consumer can employ speculative sampling or optimized

kernels for generation.

**Distinction from Semantic Communication.** Finally, we distinguish SCD from Semantic Communication or Cache-to-Cache (C2C) frameworks (Fu et al., 2026). C2C approaches typically use KV caches to fuse information from multiple agents to enhance collaborative generation. In such settings, the receiver integrates external signals into its own context. Conversely, our goal is acceleration: we enable the consumer to *skip* its prefill phase entirely. Adopting C2C-style fusion would require the consumer to first compute a local cache, negating the latency benefits of disaggregation. Thus, SCD serves as a specialized transfer protocol for latency-critical serving rather than a general collaboration mechanism.

## 3. Problem Formulation

**Disaggregated Serving and the Bandwidth Bottleneck.** Modern LLM systems disaggregate *prefill* and *decode* phases across devices to maximize utilization (Zhong et al., 2024; Qin et al., 2024). We consider a setup with a *producer* model $\mathcal{M}_A$ (Device A) and a *consumer* model $\mathcal{M}_B$ (Device B), both processing a prefix $x_{1:T}$. To generate tokens, the consumer $\mathcal{M}_B$ requires layer-wise KV caches $\mathcal{C}_B = \{(K_B^\ell, V_B^\ell)\}_{\ell=1}^L$. Device B faces a dilemma: it must either *recompute* these states locally (compute-bound) or *fetch* them from Device A (bandwidth-bound). Since the cache size $\|\mathcal{C}_B\|$ scales linearly with context length $T$ and

depth $L$, raw transmission often dominates end-to-end latency under limited interconnect bandwidth.

**Heterogeneity and Semantic Drift.** Reuse becomes challenging when producer and consumer models are heterogeneous (e.g., base vs. fine-tuned, or draft vs. verifier). Because their weights differ, their internal representations lie in distinct feature spaces. Directly substituting the producer's cache $\mathcal{C}_A$ for $\mathcal{C}_B$ causes *semantic drift*—a representation mismatch that compounds over layers. Existing methods typically mitigate this by partially recomputing specific layers, which forces a rigid trade-off between transfer cost and computation overhead (Liu et al., 2024a).

**Loss-Constrained State Transfer.** We formulate cross-model cache reuse as a rate-distortion problem. Our goal is to transfer compact *semantic codes* from $\mathcal{M}_A$ to reconstruct an *effective* cache for $\mathcal{M}_B$. We define a producer-side encoder $\phi$ and a consumer-side decoder $\psi$ operating on source states $\mathbf{S}_A$ (e.g., hidden states):

$$\mathbf{Z} = \phi(\mathbf{S}_A), \quad \mathbf{Z} \in \mathbb{R}^{T \times r}, \tag{1}$$

$$\hat{\mathbf{S}}_B = \psi(\mathbf{Z}), \tag{2}$$

where $r \ll d$ is the rank of the code, and $\hat{\mathbf{S}}_B$ is used to compute the approximated cache $\hat{\mathcal{C}}_B$.

We minimize the total transfer latency $\mathcal{T}_{\text{transfer}}$, which sums transmission, reconstruction, and residual computation time:

$$\min_{\phi, \psi} \quad \mathcal{T}_{\text{transfer}} = \frac{|\mathbf{Z}|}{\text{BW}} + \mathcal{T}_{\text{recon}}(\psi, \mathbf{Z}) + \mathcal{T}_{\text{fill}}, \tag{3}$$

where BW is the channel bandwidth, $|\mathbf{Z}|$ is the code size, and $\mathcal{T}_{\text{fill}}$ accounts for any layers explicitly recomputed rather than reconstructed (in SCD, $\mathcal{T}_{\text{fill}} = 0$ since Reuse and Patch jointly cover all layers).

Crucially, we constrain the consumer's generation quality to remain within an $\epsilon$-margin of the oracle:

$$\mathbb{E}_{x_{1:T} \sim \mathcal{D}} \Big[ D_{\text{KL}} \Big( P_{\mathcal{M}_B}(\cdot \mid \mathcal{C}_B) \,\Big\|\, P_{\mathcal{M}_B}(\cdot \mid \hat{\mathcal{C}}_B) \Big) \Big] \leq \epsilon, \tag{4}$$

where $P_{\mathcal{M}_B}$ is the next-token distribution. SCD learns layer-wise projections $(\phi, \psi)$ to minimize Eq. (3) while satisfying Eq. (4).

# 4. Method

We present **SCD**, a framework for split inference between a producer $\mathcal{M}_A$ and a consumer $\mathcal{M}_B$ that share a Transformer architecture but differ in weights. Rather than transmitting raw high-dimensional states, the producer sends compact *semantic codes*, from which the consumer reconstructs *native-space* states. SCD integrates two complementary mechanisms: (i) **Reuse**—fast low-rank reconstruction

of KV caches for the majority of layers; and (ii) **Patch**—targeted semantic correction at sparse transition layers to truncate error propagation.

## 4.1. Notation and Preliminaries

We index Transformer layers by $\ell \in \{1, \ldots, L\}$. Let $T$ denote the prefix length, $d_h$ the per-head dimension, and $d_{\text{model}}$ the model hidden size. We represent matrices in bold uppercase (e.g., $\mathbf{K}, \mathbf{V}$) and vectors/scalars in lowercase. The consumer's cache for layer $\ell$ consists of multi-head tensors $(\mathbf{K}_B^\ell, \mathbf{V}_B^\ell)$. When applying encoders/decoders, we flatten the batch $\times$ heads $\times$ tokens dimensions into a sample axis $N$. Let $\mathcal{L}_{\text{patch}} \subseteq \{1, \ldots, L\}$ be the set of layers using Patch, and $\mathcal{L}_{\text{reuse}} = \{1, \ldots, L\} \setminus \mathcal{L}_{\text{patch}}$ be the remaining layers that use Reuse. The patch budget is $k = |\mathcal{L}_{\text{patch}}|$; unless otherwise specified, we use $k=6$. We select $\mathcal{L}_{\text{patch}}$ once during offline calibration and keep it fixed for all online requests. We employ branch-specific ranks $r_K, r_V, r_H \ll d_h, d_{\text{model}}$. Keys utilize Rotary Positional Embeddings (RoPE); we denote $\text{deRoPE}(\cdot)$ and $\text{RoPE}(\cdot)$ as the inverse and forward operations. Crucially, we compress Key, Value, and Hidden states independently to preserve channel-wise semantics. Offline calibration uses a small prefix set $\mathcal{D}_{\text{cal}}$ (typically 100–500 representative prefixes). For each prefix $x_{1:T} \in \mathcal{D}_{\text{cal}}$, we run full prefills on both $\mathcal{M}_A$ and $\mathcal{M}_B$ under identical tokenization and attention masks, then record paired KV tensors and normalized pre-attention hidden states for every layer. These paired traces are used to fit Reuse translators via linear low-rank regression and Patch aligners via supervised regression before deployment. At deployment time, the learned artifacts are loaded as a static routing table keyed by layer index, and the serving path performs no per-request layer search.

## 4.2. Overview: Semantic Code Transport

As illustrated in Fig. 2, SCD replaces raw state transmission with semantic code transport. During producer prefill, we transmit codes $\{\mathbf{Z}_K^\ell, \mathbf{Z}_V^\ell\}_{\ell \in \mathcal{L}_{\text{reuse}}}$ for REUSE layers and $\{\mathbf{Z}_H^\ell\}_{\ell \in \mathcal{L}_{\text{patch}}}$ for PATCH layers. The consumer decodes $\mathbf{Z}_K/\mathbf{Z}_V$ into $\mathcal{M}_B$-native KV caches and maps $\mathbf{Z}_H$ into $\mathcal{M}_B$'s pre-attention normalized space to regenerate consistent KV pairs. Concretely, $\phi$ and $\psi$ in Eqs. (1)–(2) are instantiated as the layer-wise linear maps $\mathbf{W}_{\text{enc}}, \mathbf{W}_{\text{dec}}$ (Reuse) plus aligners $g_\theta^\ell$ (Patch).

## 4.3. Reuse: Cross-Model Low-Rank KV Reconstruction

Reuse leverages the empirical low-rank structure of KV tensors and the high subspace overlap between corresponding layers of $\mathcal{M}_A$ and $\mathcal{M}_B$. We learn a shared latent space and model-specific decoders per layer.

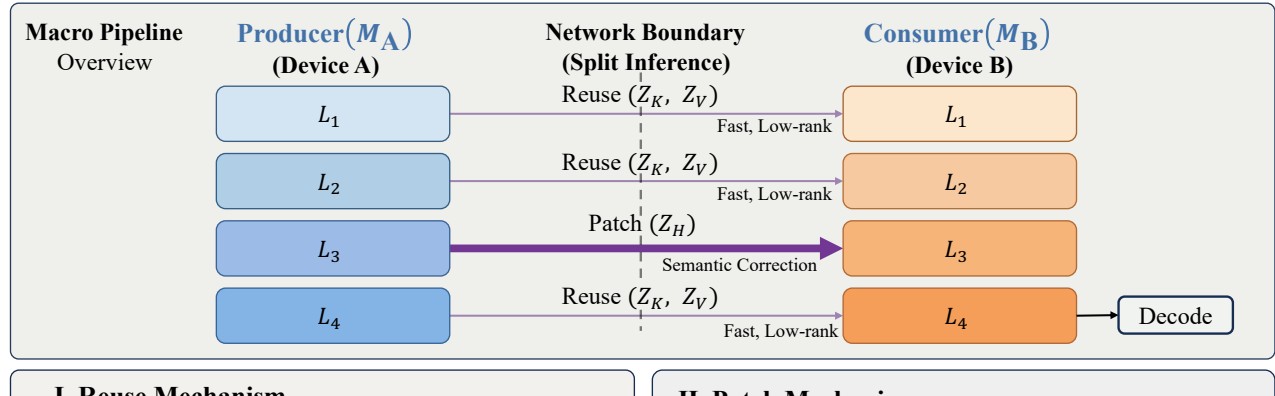

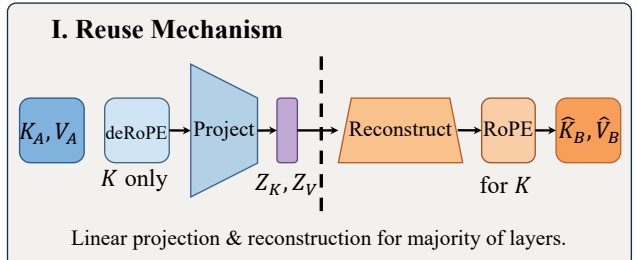
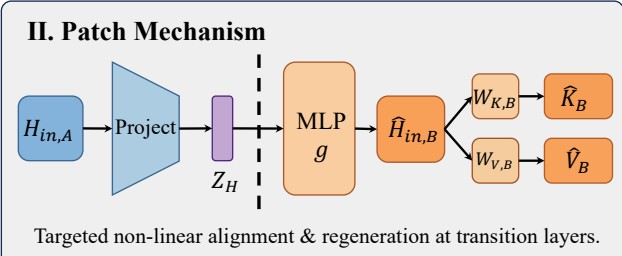

*Figure 2.* **SCD Overview.** *Offline:* We collect paired traces on identical prefixes to learn per-layer low-rank translators for KV pairs (Reuse) and aligners for transition semantic states (Patch). *Online:* The producer executes prefill, encodes, and transmits low-dimensional codes $(\mathbf{Z}_K^\ell, \mathbf{Z}_V^\ell)$ for $\ell \in \mathcal{L}_{\text{reuse}}$ and $\mathbf{Z}_H^\ell$ for $\ell \in \mathcal{L}_{\text{patch}}$. The consumer decodes these into its native-space states, bypassing local prefill.

### 4.3.1. OFFLINE: JOINT SUBSPACE LEARNING

**Paired Collection and deRoPE.** We collect per-layer tensors from paired full prefills of $\mathcal{M}_A$ and $\mathcal{M}_B$ on the same calibration prefixes $\mathcal{D}_{\text{cal}}$. The calibration stage is offline and produces frozen artifacts for a fixed model pair. For keys, we remove RoPE to operate in the content space:

$$\tilde{\mathbf{K}}_A^\ell = \text{deRoPE}(\mathbf{K}_A^\ell), \qquad \tilde{\mathbf{K}}_B^\ell = \text{deRoPE}(\mathbf{K}_B^\ell). \quad (5)$$

Values $\mathbf{V}$ are processed directly.

**Joint Low-Rank Factorization.** Flattening samples into matrices $\mathbf{A}_K^\ell, \mathbf{B}_K^\ell \in \mathbb{R}^{N \times d_h}$, we construct the joint matrix $\mathbf{X}_K^\ell = [\mathbf{A}_K^\ell \ \mathbf{B}_K^\ell] \in \mathbb{R}^{N \times 2d_h}$. A rank-$r_K$ approximation yields shared latents $\mathbf{Z}_K^\ell \in \mathbb{R}^{N \times r_K}$ and decoders $\mathbf{W}_{\text{dec},A}^\ell, \mathbf{W}_{\text{dec},B}^\ell \in \mathbb{R}^{r_K \times d_h}$ such that:

$$\mathbf{A}_K^\ell \approx \mathbf{Z}_K^\ell \mathbf{W}_{\text{dec},A}^\ell, \qquad \mathbf{B}_K^\ell \approx \mathbf{Z}_K^\ell \mathbf{W}_{\text{dec},B}^\ell. \quad (6)$$

The same procedure applies independently to values to obtain $\mathbf{Z}_V^\ell$.

**Producer Encoders.** To map new producer observations to the shared latent space, we learn a projection $\mathbf{W}_{\text{enc}}^\ell \in \mathbb{R}^{d_h \times r_K}$ via ridge regression:

$$\mathbf{W}_{\text{enc}}^\ell = \underset{\mathbf{W}}{\arg\min} \ \left\| \mathbf{A}_K^\ell \mathbf{W} - \mathbf{Z}_K^\ell \right\|_F^2 + \lambda \|\mathbf{W}\|_F^2. \quad (7)$$

### 4.3.2. ONLINE: ENCODE AND DECODE

For each $\ell \in \mathcal{L}_{\text{reuse}}$, the producer computes:

$$\mathbf{Z}_K^\ell = \tilde{\mathbf{K}}_A^\ell \mathbf{W}_{\text{enc},K}^\ell, \qquad \mathbf{Z}_V^\ell = \mathbf{V}_A^\ell \mathbf{W}_{\text{enc},V}^\ell. \quad (8)$$

The consumer reconstructs native-space caches via:

$$\tilde{\mathbf{K}}_B^\ell = \mathbf{Z}_K^\ell \mathbf{W}_{\text{dec},B}^\ell, \qquad \mathbf{V}_B^\ell = \mathbf{Z}_V^\ell \mathbf{W}_{\text{dec},B}^\ell, \quad (9)$$

followed by re-applying RoPE: $\mathbf{K}_B^\ell = \text{RoPE}(\tilde{\mathbf{K}}_B^\ell)$.

### 4.4. Patch: Semantic Correction at Transition Layers

Low-rank reconstruction is insufficient for critical layers where feature mismatches amplify. Patch intercepts the consumer's *transition semantic state* to enforce alignment.

#### 4.4.1. PATCH SIGNAL: PRE-ATTENTION NORMALIZED INPUT

For $\ell \in \mathcal{L}_{\text{patch}}$, we define the transition state as the post-norm input to the attention block:

$$\mathbf{H}_{\text{in},A}^\ell \triangleq \mathbf{x}_{\text{norm},A}^\ell, \qquad \mathbf{H}_{\text{in},B}^\ell \triangleq \mathbf{x}_{\text{norm},B}^\ell, \quad (10)$$

where $\mathbf{H} \in \mathbb{R}^{T \times d_{\text{model}}}$. The producer compresses $\mathbf{H}_{\text{in},A}^\ell$ into a code $\mathbf{Z}_H^\ell = \mathbf{H}_{\text{in},A}^\ell \mathbf{W}_{\text{enc},H}^\ell$, where $\mathbf{W}_{\text{enc},H}^\ell \in \mathbb{R}^{d_{\text{model}} \times r_H}$ is learned similarly to Eq. (7).

#### 4.4.2. CONSUMER ALIGNER AND REGENERATION

The consumer applies a non-linear aligner $g_\theta^\ell$ (an MLP) to map the code to its native space:

$$\hat{\mathbf{H}}_{\text{in},B}^\ell = g_\theta^\ell(\mathbf{Z}_H^\ell), \qquad \ell \in \mathcal{L}_{\text{patch}}. \quad (11)$$

**Algorithm 1** Split Inference with Reuse and Patch

---

**Input:** Prefix $x_{1:T}$, Producer $\mathcal{M}_A$, Consumer $\mathcal{M}_B$, fixed sets $\mathcal{L}_{\text{patch}}, \mathcal{L}_{\text{reuse}}$
**Output:** Consumer cache $\mathcal{C}_B$ ready for decoding
**Producer (Device A):**
Compute full prefill on $\mathcal{M}_A$
**for** $\ell \in \mathcal{L}_{\text{reuse}}$ **do**
    $\mathbf{Z}_K^\ell \leftarrow \text{deRoPE}(\mathbf{K}_A^\ell)\mathbf{W}_{\text{enc},K}^\ell$
    $\mathbf{Z}_V^\ell \leftarrow \mathbf{V}_A^\ell \mathbf{W}_{\text{enc},V}^\ell$
**end for**
**for** $\ell \in \mathcal{L}_{\text{patch}}$ **do**
    $\mathbf{Z}_H^\ell \leftarrow \mathbf{x}_{\text{norm},A}^\ell \mathbf{W}_{\text{enc},H}^\ell$
**end for**
Transmit $\{\mathbf{Z}_K^\ell, \mathbf{Z}_V^\ell\}_{\text{reuse}} \cup \{\mathbf{Z}_H^\ell\}_{\text{patch}}$ to Device B

---

**Consumer (Device B):**
**for** $\ell \in \mathcal{L}_{\text{reuse}}$ **do**
    $\tilde{\mathbf{K}}_B^\ell \leftarrow \mathbf{Z}_K^\ell \mathbf{W}_{\text{dec},B}^\ell$
    $\mathbf{K}_B^\ell \leftarrow \text{RoPE}(\tilde{\mathbf{K}}_B^\ell)$
    $\mathbf{V}_B^\ell \leftarrow \mathbf{Z}_V^\ell \mathbf{W}_{\text{dec},B}^\ell$
**end for**
**for** $\ell \in \mathcal{L}_{\text{patch}}$ **do**
    $\hat{\mathbf{H}}_{\text{in},B}^\ell \leftarrow g_\theta^\ell(\mathbf{Z}_H^\ell)$
    Regenerate $(\mathbf{K}_B^\ell, \mathbf{V}_B^\ell)$ via Eq. (12)
**end for**
Initiate decoding with assembled cache $\mathcal{C}_B$

---

KV pairs are then regenerated using the consumer's native weights:

$$\hat{\mathbf{K}}_B^\ell = \text{RoPE}\left(\hat{\mathbf{H}}_{\text{in},B}^\ell \mathbf{W}_{k,B}^\ell\right), \qquad (12a)$$

$$\hat{\mathbf{V}}_B^\ell = \hat{\mathbf{H}}_{\text{in},B}^\ell \mathbf{W}_{v,B}^\ell. \qquad (12b)$$

We train $g_\theta^\ell$ to minimize the L2 reconstruction error $\|g_\theta^\ell(\mathbf{Z}_H^\ell) - \mathbf{H}_{\text{in},B}^\ell\|_2^2$.

### 4.5. Online Procedure

Algorithm 1 details the inference pipeline. The transmission complexity is $O(Tr)$, significantly lower than the raw transfer cost $O(Td)$ since $r \ll d$. $\mathcal{L}_{\text{patch}}$ and $\mathcal{L}_{\text{reuse}}$ are fixed inputs from offline calibration; branch selection reduces to constant-time membership checks over layer indices, so latency variation comes from prefix length and model execution, not from an online policy search.

## 5. Theoretical Analysis

We analyze why REUSE alone leads to multiplicative error amplification across layers and how PATCH acts as a spectral truncation operator. Our analysis connects layer-wise reconstruction errors to the divergence in the consumer's output distribution. Proofs are detailed in Appendix B.

**Setup.** Consider a decoding step $t$ given prefix $x_{1:T}$. Let $\mathbf{h}_B^\ell \in \mathbb{R}^d$ denote the oracle hidden state on $\mathcal{M}_B$ at layer $\ell$ for the final prefix position (the state from which the first decode step reads), and $\hat{\mathbf{h}}_B^\ell$ denote the SCD approximation of the same state under the reconstructed cache. Let $\mathbf{o}_B, \hat{\mathbf{o}}_B \in \mathbb{R}^V$ be the corresponding output logits. We quantify the **layer-local injection errors** as:

$$\varepsilon_{\text{reuse}}^\ell \triangleq \|(\tilde{\mathbf{K}}_B^\ell, \mathbf{V}_B^\ell) - (\tilde{\mathbf{K}}_{B,\text{reuse}}^\ell, \mathbf{V}_{B,\text{reuse}}^\ell)\|, \quad (13)$$

$$\varepsilon_{\text{patch}}^\ell \triangleq \|\mathbf{h}_B^\ell - \hat{\mathbf{h}}_B^\ell\|, \quad \ell \in \mathcal{L}_{\text{patch}}, \quad (14)$$

where $\|\cdot\|$ denotes the spectral norm.

**Assumption 1: Layer-wise Stability.** We assume the Transformer layer map $\mathcal{F}_\ell$ is Lipschitz continuous with respect to both input states and cached KV pairs. For any reuse layer $\ell \in \mathcal{L}_{\text{reuse}}$, there exist stability constants $\alpha_\ell, \beta_\ell \geq 0$ such that:

$$\|\hat{\mathbf{h}}_B^{\ell+1} - \mathbf{h}_B^{\ell+1}\| \leq \alpha_\ell \|\hat{\mathbf{h}}_B^\ell - \mathbf{h}_B^\ell\| + \beta_\ell\, \varepsilon_{\text{reuse}}^\ell. \quad (15)$$

For patch layers $\ell \in \mathcal{L}_{\text{patch}}$, the aligner explicitly resets the state, bounding the error solely by the patch reconstruction quality:

$$\|\hat{\mathbf{h}}_B^\ell - \mathbf{h}_B^\ell\| \leq \varepsilon_{\text{patch}}^\ell. \quad (16)$$

**Error Propagation and Truncation.** Unrolling Eq. (15) reveals that errors accumulate multiplicatively. However, Eq. (16) introduces a reset mechanism: a Patch layer effectively truncates the history, preventing errors from earlier layers from propagating further.

**Lemma 5.1** (Segmented Error Bound). *Let patch indices be $s_1 < s_2 < \cdots < s_m$. Define segments using $s_0 \triangleq 0$ and $s_{m+1} \triangleq L$. For any target layer $\ell$ located in segment $(s_j, s_{j+1}]$, the accumulated representation error is bounded by:*

$$
\begin{aligned}
\|\hat{\mathbf{h}}_B^\ell - \mathbf{h}_B^\ell\| \leq\ & \left(\prod_{i=s_j}^{\ell-1} \alpha_i\right) \varepsilon_{\text{patch}}^{s_j} \\
& + \sum_{k=s_j}^{\ell-1}\left(\prod_{m=k+1}^{\ell-1} \alpha_m\right) \beta_k\, \varepsilon_{\text{reuse}}^k,
\end{aligned}
\quad (17)
$$

*where $\varepsilon_{\text{patch}}^{s_0} \triangleq 0$.*

*Proof Sketch.* The proof follows by induction. The base case at $\ell = s_j$ is satisfied by Eq. (16). For subsequent layers, we apply the recurrence in Eq. (15) starting from the reset point $s_j$.

**Theorem 1 (NLL Gap).** Assuming the output head is $L_{\text{out}}$-Lipschitz and the log-softmax function is locally $C_{\text{sm}}$-Lipschitz, the divergence in NLL for the target token $y_t$

satisfies:

$$|\log p(y_t) - \log \hat{p}(y_t)| \le C_{\mathrm{sm}} L_{\mathrm{out}} \|\hat{\mathbf{h}}_B^L - \mathbf{h}_B^L\|, \quad (18)$$

where the RHS is bounded by Lemma 5.1.

**Theoretical Implications.** This formalizes our *Reuse-heavy, Patch-sparse* strategy. Without Patch, the error term is dominated by $\prod_{i=1}^{L} \alpha_i$, which compounds multiplicatively with depth $L$; in practice $\alpha_i \gtrsim 1$ due to residual connections, so even small per-layer amplification accumulates into a large drift over deep stacks. By inserting a Patch at $s_j$, we replace the accumulated error term with a small local term $\varepsilon_{\mathrm{patch}}^{s_j}$, effectively truncating the drift.

# 6. Experiments

We evaluate SCD in a split inference setting where a producer runs prefill and a consumer starts decoding without a full consumer-side prefill. This setting is motivated by disaggregated serving, where KV transfer can dominate latency and requires high bandwidth at scale (Zhong et al., 2024). We focus on the practical scenario of a base model and its fine-tuned variant sharing the same architecture but different weights, for which cross-model cache reuse is known to be non-trivial (Liu et al., 2024a).

## 6.1. Experimental Setup

**Tasks and Datasets.** We evaluate generation quality and distributional fidelity across diverse benchmarks: (i) **Question Answering (QA):** We report F1 scores on CMRC2018 (Chinese extractive QA) and HotpotQA (English multi-hop reasoning), covering both span-extraction and complex reasoning tasks; (ii) **Language Modeling:** We measure Perplexity (PPL) on WikiText-2 to quantify general representational consistency; and (iii) **Distributional Fidelity:** We compute token-level KL divergence and Total Variation (TV) distance between the Oracle consumer distribution and the approximated distribution.

**Split-Inference Protocol.** Given a prefix $x_{1:T}$, the producer runs a normal prefill and transmits either raw KV (baselines) or semantic codes (SCD). The consumer reconstructs caches (and optionally applies Patch) before initiating greedy decoding.

**Calibration Protocol.** SCD uses an offline calibration set $\mathcal{D}_{\mathrm{cal}}$ only to learn auxiliary translators for a fixed model pair. For the canonical MistralLite→Mistral-7B run, we use 200 CMRC2018 prefixes (mean prefix length 609.0 tokens; 121,805 total tokens). For each prefix, we run full prefills on both producer and consumer with identical tokenization and attention masks, then save paired per-layer KV tensors and

normalized pre-attention hidden states. The REUSE module is fitted by low-rank subspace identification followed by ridge-regression encoders/decoders; the PATCH module trains sparse MLP aligners on the selected transition layers. The learned artifacts are frozen and reused across online requests.

**Bandwidth Measurement.** We measure TTFT and end-to-end latency under controlled effective bandwidth $B_{\mathrm{net}}$ (sweeping from 100 Gbps to 1 Tbps). We report the total transmitted payload size for each method.

**Method Compared.** We compare the following methods:

- **Oracle**: Full prefill on $\mathcal{M}_B$ (Upper bound on quality; Lower bound on latency).

- **Raw KV**: Transmits full BF16 KV caches from producer to consumer.

- **Quantized KV**: Transmits **4-bit quantized** KV caches (group-wise INT4) and dequantizes on the consumer.

- **DroidSpeak-style Selective Recompute** (Liu et al., 2024a): Reuses raw KV for most layers but fully recomputes a selected subset of critical layers on $\mathcal{M}_B$, following the selection policy proposed in DroidSpeak.

- **SCD (Reuse-only)**: Applies low-rank reuse to all layers (i.e., $\mathcal{L}_{\mathrm{patch}} = \emptyset$); no recomputation is performed.

- **SCD**: The proposed full method, applying Patch at selected transition layers and Reuse elsewhere.

**Implementation Details.** We use the same tokenization and exact attention masking on both sides. Keys are deRoPE'd before encoding and reRoPE'd after decoding. We keep $K/V/H$ separate with independent ranks as described in Section 4 and Appendix F. Offline calibration cost and deployment footprint are reported in Section 6.6.

## 6.2. Main Results

### 6.2.1. BANDWIDTH–LATENCY TRADE-OFF

**Bandwidth–Latency Analysis.** Figure 3 sweeps the link bandwidth from 100 Gbps to 1 Tbps. As bandwidth increases, TTFT decreases and asymptotically approaches the compute/synchronization floor. **Quantized KV (4-bit)** achieves the lowest TTFT, consistent with its minimal payload size. However, as shown in Table 2, this comes at the cost of unusable generation quality. **SCD (Reuse+Patch)** improves significantly over **DroidSpeak-style Selective Recompute**, with its overhead dominated by local reconstruction (GEMMs) rather than data transfer. This confirms SCD's efficiency in bandwidth-constrained regimes typical of disaggregated serving.

*Table 1.* **Main Results across Model Scales.** Mistral (7B) and Qwen (32B) pairs on the QA benchmarks of Section 6.1. **SCD** consistently recovers Oracle-level quality across scales while achieving ∼2.0× speedup over full prefill. Oracle TTFT differs across tables due to per-dataset prefix length distributions; underlying model pairs are unchanged.

| | **Mistral Pair (7B)** | | | | **Qwen Pair (32B)** | | | |
|---|---|---|---|---|---|---|---|---|
| **Method** | **TTFT** (ms) | **Speedup** (vs. Oracle) | **F1** (QA) | **KL** | **TTFT** (ms) | **Speedup** (vs. Oracle) | **F1** (QA) | **KL** |
| Oracle | 237.1 | 1.00× | 0.807 | 0.00 | 503.6 | 1.00× | 0.876 | 0.00 |
| Raw KV (BF16) | 61.3 | 3.87× | 0.192 | 4.68 | 128.7 | 3.91× | 0.283 | 8.16 |
| Quantized (4-bit) | 54.6 | 4.34× | 0.151 | 7.16 | 115.3 | 4.37× | 0.212 | 10.27 |
| DroidSpeak-style | 139.4 | 1.70× | 0.754 | 0.31 | 278.4 | 1.81× | 0.813 | 0.49 |
| **SCD** | 119.7 | **1.98×** | **0.769** | **0.26** | 243.7 | **2.07×** | **0.847** | **0.31** |

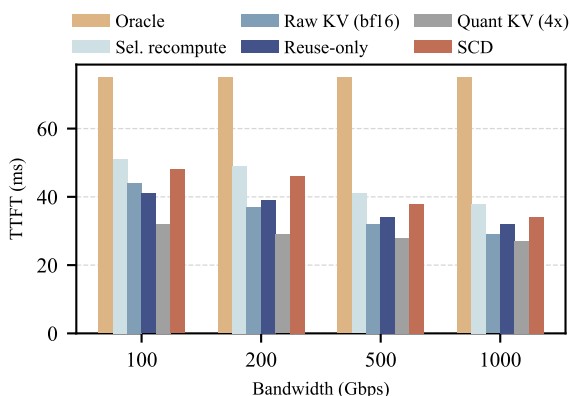

*Figure 3.* **Bandwidth–Latency Trade-off.** Time-to-first-token (TTFT) versus effective bandwidth (log-scale). While **Quantized KV (4-bit)** is the fastest due to minimal payload, it suffers from catastrophic quality collapse (see Table 2). **SCD** achieves a favorable sweet spot: it is significantly faster than Raw KV and DroidSpeak-style Selective Recompute while maintaining Oracle-level quality.

*Table 2.* **Main Results: Quality vs. Efficiency.** Evaluated on MistralLite→Mistral-7B at $B_{net}$=200 Gbps. **SCD** matches Oracle quality (F1 0.78 vs 0.81) while achieving **2.65×** speedup. Note that **4-bit Quantization** fails to bridge the cross-model semantic gap (F1 0.13).

| Method | Payload (MB/req) | TTFT (ms) | Speedup (vs. Oracle) | F1 Score (CMRC) | KL Div. (WikiText) | PPL (WikiText) |
|---|---|---|---|---|---|---|
| Oracle | 0.00 | 331.9 | 1.00× | 0.8105 | 0.0000 | 4.15 |
| Raw KV (BF16) | 115.02 | 145.5 | 2.28× | 0.2725 | 3.1209 | >100 |
| Quantized (4-bit) | **28.76** | **59.2** | **5.60×** | 0.1289 | 6.1887 | >100 |
| DroidSpeak-style | 127.34 | 198.7 | 1.67× | 0.7052 | 0.1459 | 5.33 |
| SCD (Reuse-only) | 69.02 | 118.2 | 2.81× | 0.6421 | 0.1661 | 6.11 |
| **SCD** | 87.31 | 125.2 | **2.65×** | **0.7850** | **0.1105** | **4.65** |

*Table 3.* **Effect of Patch Budget ($k$) on WikiText-2.** Increasing the number of patched layers ($k$) monotonically improves perplexity (PPL) at the cost of modest TTFT overhead. The default $k$=6 used in the main experiments is chosen from the F1-based efficiency curve in Figure 5 rather than this PPL profile, which is why $k$=6 is not listed explicitly in this table.

| Configuration | Patched Layers | TTFT (ms) | PPL | $\Delta$ PPL |
|---|---|---|---|---|
| Reuse-only | 0 | 112.3 | 6.788 | 0.00 |
| + Patch $\ell_0$ | 1 | 115.9 | 5.971 | -0.82 |
| + Patch Top-3 | 3 | 125.7 | 5.326 | -1.46 |
| + Patch Top-5 | 5 | 136.4 | 5.017 | -1.77 |
| + Patch Top-8 | 8 | 145.9 | 4.816 | -1.97 |

### 6.2.2. QUALITY AND FIDELITY

Table 1 and Table 2 highlight the necessity of semantic-aware transfer. **Quantized KV (4-bit)** collapses quality (F1 0.1289), proving that standard compression cannot handle the weight mismatch between heterogeneous models. **Reuse-only** (all-layer reuse) improves F1 to 0.6421 but still lags behind Oracle. By adding **Patch**, SCD recovers nearly full Oracle performance (F1 0.7850) with only a modest latency increase (+7.0 ms), effectively bridging the semantic gap. Crucially, Table 1 demonstrates that this advantage holds for the larger **Qwen-32B** pair, validating the scalability of our approach. Additional adapted compression baselines are reported in Appendix E.

### 6.3. Ablations

#### 6.3.1. RANK SENSITIVITY AND PARETO FRONTIER

Figure 4 plots the quality–cost trade-off. The Pareto frontier demonstrates that SCD offers tunable compression. We observe that $r_K$ can be reduced more aggressively than

$r_V$ without hurting quality, suggesting that Value states carry more fine-grained semantic information required for generation.

#### 6.3.2. PATCH NECESSITY AND BUDGET

Table 3 confirms that PATCH is not cosmetic. Starting from REUSE-ONLY (PPL 6.788), adding just 5 patched layers reduces PPL to 5.017, close to the Oracle baseline. This supports our hypothesis that error propagation is concentrated in a few critical layers; correcting these via Patch stabilizes the entire generation process.

### 6.4. Analysis: Error Propagation

Figure 6 visualizes the truncation effect predicted by our theory. With REUSE-ONLY, feature mismatch compounds with depth. In contrast, SCD resets the error at each patched layer, keeping the trajectory close to the Oracle manifold.

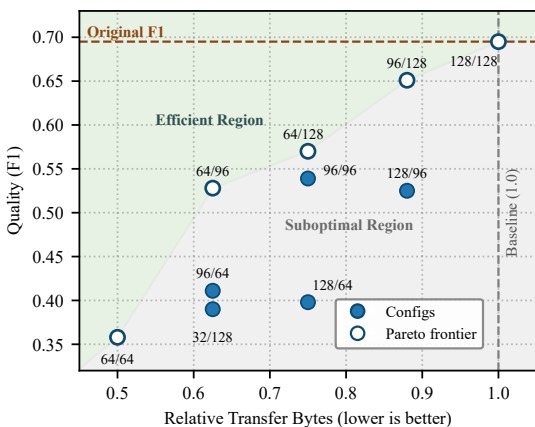

*Figure 4.* **Rank Sensitivity (Pareto Frontier).** Quality (F1) versus transferred bytes across $(r_K, r_V)$ configurations. Hollow markers denote non-dominated operating points. Reducing $r_K$ yields substantial compression with minimal quality loss, highlighting the asymmetric information density in K vs. V.

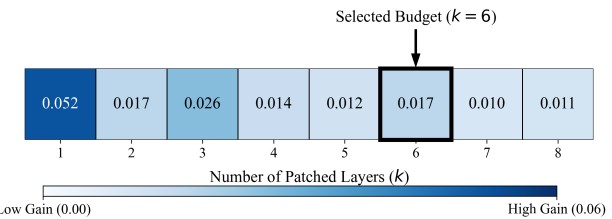

*Figure 5.* **Patch Budget Selection.** Cumulative efficiency $\Delta F(\mathcal{S}_k)/(c \cdot k)$ as a function of the patch budget $k$, where $\Delta F$ is the F1 gain over the all-Reuse baseline and $c$ is the per-layer patch cost (Appendix A). We select $k=6$ as the default budget at the argmax of this curve; smaller $k$ leaves quality on the table, while larger $k$ adds cost faster than it adds quality.

### 6.5. End-to-End Latency Breakdown

Table 4 dissects the latency. **Reuse-only** eliminates the recomputation overhead (0.0 ms), achieving the fastest total time among reuse-based methods (112.9 ms), though at lower quality. **SCD** introduces a modest 11.7 ms overhead in the "Recompute" phase, which corresponds to the execution of the Patch Aligner networks and the regeneration of native KV pairs at transition layers. This small investment yields the quality jump observed in Table 2.

### 6.6. Offline Calibration and Deployment Cost

SCD shifts cross-model adaptation into a bounded offline phase. For each producer–consumer pair, we collect paired traces, fit the Reuse translators, profile layer sensitivity, and train Patch aligners for the selected transition layers. For the canonical MistralLite→Mistral-7B run, the single-GPU pipeline takes 9.74 hours total, dominated by Patch aligner training; Reuse fitting is lightweight and neither stage retrains either backbone (per-stage breakdown in Appendix F, Table 6). The learned artifacts are reusable across requests

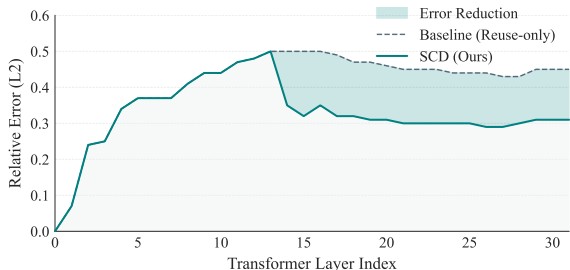

*Figure 6.* **Layer-wise Error Propagation.** Relative $\ell_2$ error of the pre-attention normalized input $x_{\mathrm{norm},B}^{\ell}$. REUSE-ONLY exhibits compounding error accumulation. SCD (Patch) effectively truncates this error at transition layers (marked by dashed lines), preventing downstream drift.

*Table 4.* **End-to-end TTFT Breakdown (WikiText-2).** SCD achieves a **2.19×** speedup over Oracle (271.3 ms) with minimal reconstruction overhead. Column abbreviations: **Prod.** (Producer Prefill), **Enc.** (Encoding), **Trans.** (Network Transfer), **Reb.** (Rebuild/Decompression), **Rec.** (Recomputation/Patching), **Dec₁** (First Token Decode).

| Method | Prod. (ms) | Enc. (ms) | Trans. (ms) | Reb. (ms) | Rec. (ms) | Dec₁ (ms) | Total (ms) | Speedup (vs. Oracle) |
|---|---|---|---|---|---|---|---|---|
| Oracle | – | – | – | – | – | – | 271.3 | 1.00× |
| DroidSpeak-style | 78.1 | 0.0 | 10.7 | 1.4 | 23.0 | 20.7 | 133.9 | 2.03× |
| Reuse-only | 78.4 | 2.6 | 7.9 | 2.7 | **0.0** | 21.3 | **112.9** | **2.40×** |
| SCD | 77.6 | 3.1 | 8.5 | 2.1 | 11.7 | 21.1 | 124.1 | 2.19× |

for the same model pair: 8.4 MB for Reuse, 1.58 GB for Patch, totaling 397.7M auxiliary parameters (5.49% of the 7.24B consumer). Thus calibration is a one-time per-pair expense, and serving loads these frozen artifacts and follows the offline-selected route.

## 7. Conclusion

We proposed Semantic Cache Distillation (SCD), a state transfer framework for disaggregated LLM serving that replaces opaque KV transfer with compact semantic codes. By combining REUSE (low-rank reconstruction) with PATCH (sparse semantic correction), SCD bridges weight-mismatched feature spaces, truncates error propagation, and recovers Oracle-level quality with substantially lower communication overhead.

**Limitations and Future Work.** SCD targets shared-architecture, weight-mismatched model pairs; arbitrary cross-architecture transfer, extreme long-context regimes beyond 32K tokens, and large distribution shifts remain open. Each pair also requires one-time calibration; future work will address these boundaries with bandwidth-adaptive ranks and broader transfer.

## Impact Statement

This work focuses on the efficiency of LLM inference, particularly in disaggregated serving scenarios. By mitigating bandwidth overheads for state transfer between heterogeneous models, our approach reduces the energy consumption and operational costs associated with large-scale AI services. This contributes to the sustainability and accessibility of AI infrastructure. We do not foresee specific negative ethical or societal consequences intrinsic to this transport protocol beyond the general risks associated with LLM deployment.

## Acknowledgements

This work was supported in part by National Natural Science Foundation of China (NSFC) under Grant 62272050 and Grant 62302048; in part by the Guangdong Key Lab of AI and Multi-modal Data Processing, Beijing Normal-Hong Kong Baptist University (BNBU), Zhuhai under 2023-2024 Grants sponsored by Guangdong Provincial Department of Educationin part by Institute of Artificial Intelligence and Future Networks (BNU-Zhuhai) and Engineering Center of AI and Future Education, Guangdong Provincial Department of Science and Technology, ChinaZhuhai Science-Tech Innovation Bureau under Grant No. 2320004002772, and in part by the Interdisciplinary Intelligence SuperComputer Center of Beijing Normal University (Zhuhai).

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

## A. Layer Policy: Selecting Reuse vs. Patch

SCD applies REUSE to the majority of layers to minimize transfer overhead, while enabling PATCH on a sparse subset to recover generation quality. Let $\mathcal{L} = \{0, \dots, L-1\}$ be the set of Transformer layers. Let $\mathcal{S} \subseteq \mathcal{L}$ denote the set of layers where the consumer applies PATCH (during profiling, this corresponds to restoring oracle states), with the remaining layers $\mathcal{L} \setminus \mathcal{S}$ using REUSE. Let $F(\mathcal{S})$ be a task metric (e.g., F1 score or negative log-likelihood) measured under this configuration. The marginal gain over the all-REUSE baseline is:

$$\Delta F(\mathcal{S}) \triangleq F(\mathcal{S}) - F(\emptyset). \tag{19}$$

Assuming a constant per-layer cost $c$ (e.g., latency or parameters) for patching, we select $\mathcal{S}$ to maximize efficiency:

$$\text{Eff}(\mathcal{S}) \triangleq \frac{\Delta F(\mathcal{S})}{c \cdot |\mathcal{S}|}. \tag{20}$$

**Step 1: Restore-one Sensitivity (Candidate Discovery).**   Starting from the all-REUSE baseline ($\mathcal{S} = \emptyset$), we measure the individual marginal benefit of restoring each layer $\ell$:

$$\Delta F_\ell \triangleq F(\{\ell\}) - F(\emptyset), \qquad \ell \in \mathcal{L}. \tag{21}$$

We construct a candidate pool $\mathcal{C}$ by selecting the top-$M$ layers ranked by $\Delta F_\ell$.

**Step 2: Interaction-aware Greedy Selection.**   Single-layer sensitivity ignores inter-layer interactions. To address this, we construct $\mathcal{S}$ via forward greedy selection on the candidate pool $\mathcal{C}$:

$$\ell_t = \underset{\ell \in \mathcal{C} \setminus \mathcal{S}_{t-1}}{\arg\max} \Big( F(\mathcal{S}_{t-1} \cup \{\ell\}) - F(\mathcal{S}_{t-1}) \Big), \quad \mathcal{S}_t = \mathcal{S}_{t-1} \cup \{\ell_t\}. \tag{22}$$

Unlike approaches that restrict recomputation to contiguous segments, SCD allows $\mathcal{S}_t$ to be sparse. This flexible topology enables PATCH to act as a semantic bridge at critical transitions without requiring dense support.

**Step 3: Budgeted Cost-aware Selection.**   Instead of fixing the budget a priori, we determine the optimal size $k^\star \in [k_{\min}, k_{\max}]$ that maximizes the efficiency metric:

$$k^\star = \underset{k \in [k_{\min}, k_{\max}]}{\arg\max} \frac{\Delta F(\mathcal{S}_k)}{c \cdot k}. \tag{23}$$

The set $\mathcal{S}_{k^\star}$ defines the PATCH policy; all other layers employ REUSE.

**Analysis of Layer Selection Dynamics.**   We validate our policy by analyzing the sensitivity and interactions of PATCH layers, as shown in Figure 7. (a) **Sparsity:** The sensitivity profile shows a heavy-tailed distribution. A small subset of critical layers drives most of the generation quality, while the majority have negligible impact. (b) **Redundancy:** Greedy selection shows diminishing returns. The cumulative gain plateaus as the budget increases, indicating significant redundancy among lower-ranked layers and suggesting that dense restoration is wasteful. (c) **Efficiency:** Maximizing the gain-per-cost ratio identifies an optimal operating point at a small $k$ (green marker). This confirms our design choice to apply PATCH as a sparse, targeted intervention rather than a contiguous block.

## B. Proofs for Section 5

This appendix provides proofs for Lemma 5.1 and Eq. (18), using the notation from the main text.

### B.1. Useful Inequalities for Log-Softmax

Let $\text{lse}(\mathbf{o}) \triangleq \log \sum_i \exp(o_i)$ denote the LogSumExp function. The log-softmax vector is $\log \text{softmax}(\mathbf{o})_y = o_y - \text{lse}(\mathbf{o})$.
**Lemma B.1** (Stability of Log-Softmax). *For any logits $\mathbf{o}, \hat{\mathbf{o}} \in \mathbb{R}^V$ and token index $y$,*

$$\left| - \log \text{softmax}(\hat{\mathbf{o}})_y + \log \text{softmax}(\mathbf{o})_y \right| \leq 2 \|\hat{\mathbf{o}} - \mathbf{o}\|_\infty. \tag{24}$$

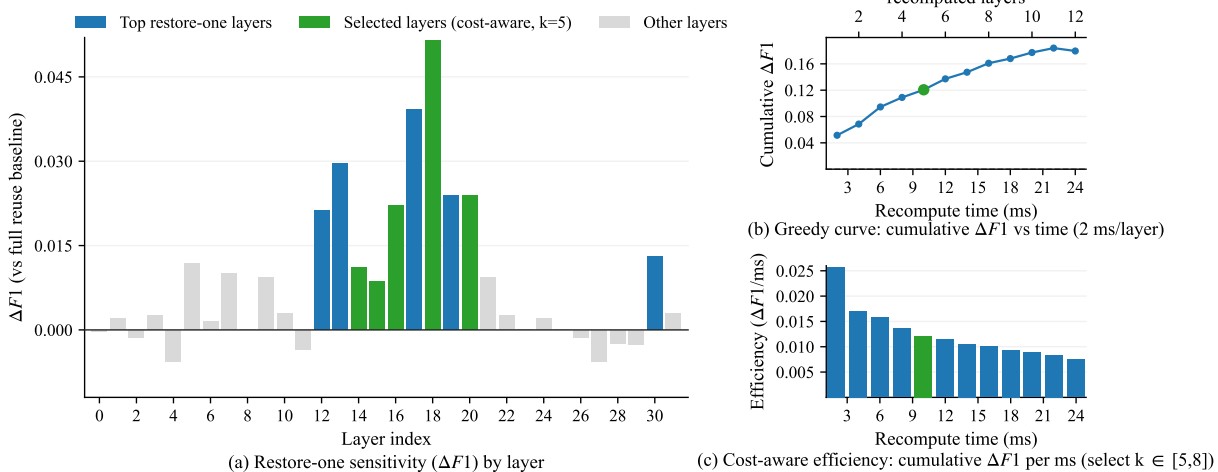

(b) Greedy curve: cumulative $\Delta F1$ vs time (2 ms/layer)

(a) Restore-one sensitivity ($\Delta F1$) by layer

(c) Cost-aware efficiency: cumulative $\Delta F1$ per ms (select $k \in [5,8]$)

*Figure 7.* **Visualization of the Layer Selection Policy. (a) Candidate Discovery:** The restore-one sensitivity profile reveals that only a few critical layers (blue) yield significant quality gains. **(b) Interaction Effects:** Greedy selection shows diminishing returns, indicating that restoring dense contiguous segments incurs high redundancy. **(c) Budget Optimization:** The efficiency curve $\Delta F(\mathcal{S}_k)/k$ peaks at a sparse budget $k^\star$ (green marker), representing the optimal trade-off between generation quality and computational cost.

*Proof.* Decompose the difference as:
$$\Delta = (o_y - \hat{o}_y) + (\mathrm{lse}(\hat{\mathbf{o}}) - \mathrm{lse}(\mathbf{o})).$$

The first term is bounded by $\|\mathbf{o} - \hat{\mathbf{o}}\|_\infty$. For the second term, observe that $\mathrm{lse}(\cdot)$ is 1-Lipschitz with respect to the $\|\cdot\|_\infty$ norm, since its gradient is $\mathrm{softmax}(\cdot)$, which has an $\ell_1$ norm of exactly 1. Thus, $|\mathrm{lse}(\hat{\mathbf{o}}) - \mathrm{lse}(\mathbf{o})| \leq \|\hat{\mathbf{o}} - \mathbf{o}\|_\infty$. Summing the bounds yields Eq. (24). $\qquad\square$

## B.2. Proof of Lemma 5.1

*Proof.* Let the sorted patch layers be $\mathcal{L}_{\mathrm{patch}} = \{s_1 < \cdots < s_m\}$, with sentinels $s_0 = 0$ and $s_{m+1} = L$. Fix a target layer $\ell \in (s_j, s_{j+1}]$.

**Initialization (Segment Start).** If $j = 0$, we have $\hat{\mathbf{h}}^{s_0} = \mathbf{h}^{s_0}$, so $\|\hat{\mathbf{h}}^{s_0} - \mathbf{h}^{s_0}\| = 0$. If $j \geq 1$, by the Patch injection bound (Eq. (16)), the error is reset:
$$\|\hat{\mathbf{h}}^{s_j} - \mathbf{h}^{s_j}\| \leq \varepsilon_{\mathrm{patch}}^{s_j}.$$

**Unrolling (Within Segment).** For any layer $u \in \{s_j, \ldots, \ell - 1\}$ utilizing Reuse, we apply the stability condition (Eq. (15)):
$$\|\hat{\mathbf{h}}^{u+1} - \mathbf{h}^{u+1}\| \leq \alpha_u \|\hat{\mathbf{h}}^u - \mathbf{h}^u\| + \beta_u \varepsilon_{\mathrm{reuse}}^u.$$

Unrolling this recursion from $u = s_j$ to $u = \ell - 1$ yields:
$$\|\hat{\mathbf{h}}^\ell - \mathbf{h}^\ell\| \leq \left(\prod_{i=s_j}^{\ell-1} \alpha_i\right) \|\hat{\mathbf{h}}^{s_j} - \mathbf{h}^{s_j}\| + \sum_{k=s_j}^{\ell-1} \left(\prod_{m=k+1}^{\ell-1} \alpha_m\right) \beta_k \varepsilon_{\mathrm{reuse}}^k.$$

Substituting the initialization bound $\varepsilon_{\mathrm{patch}}^{s_j}$ completes the proof. $\qquad\square$

## B.3. Proof of Eq. (18) (NLL Gap)

*Proof.* Assume the output head (LayerNorm + Linear) is $L_{\mathrm{out}}$-Lipschitz under the infinity norm:
$$\|\hat{\mathbf{o}} - \mathbf{o}\|_\infty \leq L_{\mathrm{out}} \|\hat{\mathbf{h}}^L - \mathbf{h}^L\|.$$

By Lemma B.1, with $\hat{\mathbf{o}} = \hat{\mathbf{o}}_B$ and $\mathbf{o} = \mathbf{o}_B$:
$$|-\log \hat{p}(y_t) + \log p(y_t)| \leq 2\|\hat{\mathbf{o}}_B - \mathbf{o}_B\|_\infty \leq 2L_{\mathrm{out}} \|\hat{\mathbf{h}}^L - \mathbf{h}^L\|.$$

This corresponds to Eq. (18) (setting $C_{sm} = 2$). Substituting the bound for $\|\hat{\mathbf{h}}^L - \mathbf{h}^L\|$ from Lemma 5.1 yields the final result. $\qquad\square$

# C. Estimating Stability Constants $\alpha_\ell$ and $\beta_\ell$

We provide a concrete instantiation of the stability coefficients $\alpha_\ell$ (state sensitivity) and $\beta_\ell$ (cache sensitivity). While standard attention is not globally Lipschitz, we utilize *local* Lipschitz constants estimated on a bounded domain defined by the calibration set, consistent with prior literature (Castin et al., 2023; Yudin et al., 2025).

## C.1. Layer Map and Perturbation

Consider a pre-LN decoder layer (omitting the layer index $\ell$ for brevity). The forward pass is defined as:

$$\mathbf{u} = \text{LN}(\mathbf{h}), \tag{25}$$

$$\mathbf{Q} = \mathbf{u}\mathbf{W}_Q, \quad \mathbf{K}, \mathbf{V} \in \mathbb{R}^{T \times d_h}, \tag{26}$$

$$\mathbf{A} = \text{softmax}\left(\frac{\mathbf{Q}\mathbf{K}^\top}{\sqrt{d_h}} + \mathbf{M}\right), \tag{27}$$

$$\text{Attn}(\mathbf{h}; \mathbf{K}, \mathbf{V}) = (\mathbf{A}\mathbf{V})\mathbf{W}_O. \tag{28}$$

Our goal is to bound the output perturbation $\|\delta\mathbf{h}^{\ell+1}\|$ in terms of the input error $\|\delta\mathbf{h}^\ell\|$ and cache errors ($\|\delta\mathbf{K}\|, \|\delta\mathbf{V}\|$).

## C.2. Component Bounds

**Softmax Jacobian.** For the row-wise softmax function, the spectral norm of the Jacobian is bounded by $1/2$. Locally, we have:

$$\|\delta\mathbf{A}\|_2 \leq \frac{1}{2}\|\delta\mathbf{S}\|_2, \tag{29}$$

where $\mathbf{S}$ denotes the pre-softmax scores.

**LayerNorm.** We denote the local Lipschitz constant of LayerNorm as $L_{\text{LN}}^\ell$. We estimate this empirically based on the minimum per-token variance observed in the calibration traces.

## C.3. Bounding Attention and MLP

By expanding the differential of the attention mechanism and applying triangle inequalities (following the derivation in Kim et al. (2021)), we derive the aggregate stability coefficients:

$$\alpha_\ell := 1 + L_{\text{attn},h}^\ell + L_{\text{mlp}}^\ell, \tag{30}$$

$$\beta_\ell := L_{\text{attn},K}^\ell + L_{\text{attn},V}^\ell. \tag{31}$$

The term $\alpha_\ell$ includes 1 due to the residual connection. The cache sensitivity terms are defined as:

- $L_{\text{attn},K}^\ell$: Captures sensitivity to Key perturbations, mediated by the query magnitude $\|\mathbf{Q}\|_2$.

- $L_{\text{attn},V}^\ell$: Captures sensitivity to Value perturbations, mediated by the attention probability norm $\|\mathbf{A}\|_2$.

## C.4. Practical Estimation

We estimate these constants using statistics collected from the calibration set:

- **Weights:** We compute spectral norms $\|\mathbf{W}\|_2$ via power iteration.

- **Activations:** We use the empirical maxima of $\|\mathbf{K}\|_2, \|\mathbf{V}\|_2, \|\mathbf{Q}\|_2$ over the traces to bound the local domain.

- **Softmax:** We use the theoretical bound $L_{\text{sm}} = 1/2$.

This procedure yields tight layer-wise coefficients $(\alpha_\ell, \beta_\ell)$ that accurately predict the error amplification observed in our experiments.

## D. Additional Visualizations

**Visualization of Representation Alignment.** Figure 8 visualizes KV representations using t-SNE to assess alignment quality. We compare source (producer, blue), target (consumer, gray), and translated (orange) states. The raw source distribution is clearly separated from the target, indicating a significant feature gap. In contrast, the translated states align closely with the target clusters. This demonstrates that our method effectively maps producer states into the consumer's semantic space.

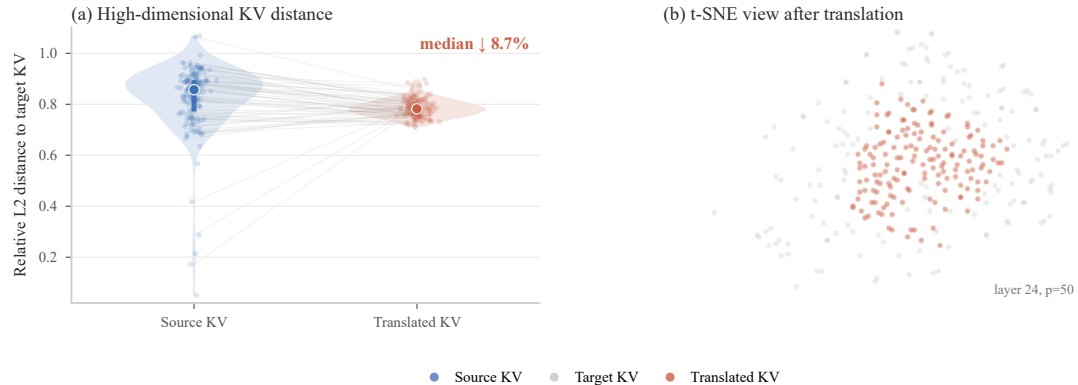

*Figure 8.* **t-SNE visualization of cross-model KV alignment.** Blue, gray, and orange points represent source, target, and translated KV states, respectively. The translated states move closer to the target distribution, illustrating the reduction of the representational gap.

## E. Neighboring Compression Baselines

We additionally adapt representative neighboring compression methods, including SVD-LLM (Wang et al., 2025a), SliceGPT (Ashkboos et al., 2024), and TensorLLM (Gu et al., 2025), to the heterogeneous state-transfer setting. These methods primarily target model or tensor compression rather than producer-to-consumer KV transfer under weight mismatch, so we report them in the appendix instead of the main comparison. Table 5 shows that generic compression alone does not resolve cross-model semantic misalignment.

*Table 5.* **Adapted Neighboring Compression Baselines.** Quality retention and TTFT speedup are measured under the same heterogeneous transfer setting. Values above $1.0\times$ indicate speedup over the oracle consumer prefill; values below $1.0\times$ indicate slowdown.

| Method | Quality Retention | TTFT Speedup |
|---|---|---|
| SVD-LLM-style adaptation | 70.6% | $0.83\times$ |
| SliceGPT-style adapted KV transfer | 85.0% | $0.95\times$ |
| TensorLLM-style adapted KV transfer | 0.0% | $0.16\times$ |
| **SCD** | 95–97% | $1.98$–$2.65\times$ |

The results support the distinction between compression and semantic transfer. SVD-LLM-style and SliceGPT-style adaptations retain part of the consumer behavior but do not provide a latency benefit in this split-inference setting. TensorLLM-style adaptation is more aggressive but collapses quality under weight mismatch. SCD instead learns producer-to-consumer translators and applies Patch at sparse transition layers, preserving most oracle quality while improving TTFT.

## F. Extended Implementation Details

We divide our implementation into two phases: an **Offline Calibration Phase**, where we learn layer-wise parameters to bridge model discrepancies; and an **Online Execution Phase**, where the target model ($\mathcal{M}_B$) uses these parameters to populate its KV cache, regenerating native KV pairs at selected patch layers. A core principle is to follow the *native KV-cache semantics* of the inference framework (e.g., Hugging Face Transformers), ensuring that reconstructed states are treated as standard cached history during generation.

## F.1. Offline Calibration and Parameter Training

**Reuse: Layer-wise Stack-SVD with Ridge Regression.** We implement REUSE as a stack of independent linear translators. For each layer $\ell$, we execute full prefill passes on both the source ($\mathcal{M}_A$) and target ($\mathcal{M}_B$) models over a calibration dataset. We collect the internal representations at layer $\ell$ that serve as inputs to the KV cache. The optimization proceeds in two steps:

1. **Subspace Identification via Truncated SVD:** We first apply Truncated Singular Value Decomposition (SVD) to identify a rank-$r$ latent subspace. By the Eckart–Young–Mirsky theorem, this yields the optimal low-rank approximation of the feature map in the Frobenius norm, minimizing information loss.
2. **Mapping Fit via Ridge Regression:** We fit the linear encoder/decoder mappings using Ridge regression ($\ell_2$ regularization). We choose Ridge over unregularized least squares to improve numerical stability and prevent noise amplification from small singular values.

**Patch: Sparse Breakpoint Alignment.** We train the PATCH mechanism only on a sparse subset of transition layers $\ell \in \mathcal{L}_{\text{patch}}$. For each layer, we collect the "pre-attention breakpoint state" (the state immediately preceding the self-attention projection). We train an aligner $g^\ell$ to map the low-dimensional code from $\mathcal{M}_A$ to an estimate of $\mathcal{M}_B$'s state. The training objective is a standard regression loss (e.g., MSE). To improve consistency, we can optionally include auxiliary terms matching intermediate representations, similar to feature matching in Knowledge Distillation. The patch set $\mathcal{L}_{\text{patch}}$ is selected offline under the budget $k$ and is loaded as a fixed route at serving time.

The final offline artifacts are:

- **Reuse Artifacts:** Linear encoder/decoder parameters for all layers.
- **Patch Artifacts:** Breakpoint aligners $g^\ell$ for the subset $\mathcal{L}_{\text{patch}}$.

These are serialized with minimal metadata for loading at runtime.

*Table 6.* **Offline Calibration Cost.** One-time cost for MistralLite→Mistral-7B using 200 CMRC2018 prefixes on a single GPU.

| Stage | Time (s) | Peak Mem. (GB) |
|---|---|---|
| Paired trace collection | 1,068.9 | 18.1 |
| Reuse fitting | 905.5 | 3.0 |
| Layer-wise profiling | 2,622.2 | 9.4 |
| Patch aligner training | 30,456.9 | 7.7 |
| Patch resharding | 11.9 | – |
| **Total** | **9.74 h** | **18.1** |

## F.2. Online Execution and Hybrid Cache Construction

**Online Reuse: Transmission and Reconstruction.** During inference, $\mathcal{M}_A$ performs a standard prefill and transmits low-dimensional codes for each reused layer $\ell$. $\mathcal{M}_B$ uses the corresponding decoder to reconstruct the tensor in its own feature space and writes it directly into its cache container (e.g., `past_key_values`). Once the cache is populated, $\mathcal{M}_B$ skips its local prefill and enters the standard decoding loop. The framework handles cache indices (e.g., `cache_position`) natively, treating the reconstructed cache as valid local history.

**Online Patch: Alignment and Recomputation.** When the PATCH mechanism is applied at a selected transition layer $\ell$, the execution flow is:

1. **State Alignment:** $\mathcal{M}_B$ uses the aligner $g^\ell$ to map the incoming code to an estimated normalized pre-attention state in the consumer space.
2. **Native KV Regeneration:** $\mathcal{M}_B$ multiplies this aligned state by its own $W_{k,B}^\ell$ and $W_{v,B}^\ell$, and applies RoPE to the key, producing consumer-native KV pairs for layer $\ell$.
3. **Cache Assembly:** The final cache is a hybrid: layers in $\mathcal{L}_{\text{reuse}}$ use reconstructed KV pairs, and layers in $\mathcal{L}_{\text{patch}}$ use patch-regenerated native KV pairs.

