# OpenReview forum: "Semantic Cache Distillation: Efficient State Transfer via Reuse and Selective Patching"
_ICML.cc/2026/Conference — ICML 2026 regular_

### Official Review · Reviewer_zFYi · 2026-03-02

**Soundness:** 2
**Presentation:** 2
**Significance:** 2
**Originality:** 2
**Overall Recommendation:** 3
**Confidence:** 3

**Summary:**

This paper introduces Semantic Cache Distillation (SCD) for efficient KV cache transfer in disaggregated LLM serving with heterogeneous models. SCD transmits low-rank semantic codes and reconstructs consumer-aligned states using a combination of Reuse and Patch. The results show that the proposed scheme achieve better bandwidth and latency while maintaining near-oracle generation quality.

**Compliance With Llm Reviewing Policy:**

Affirmed.

**Final Justification:**

The response partially resolved my concerns, but questions remain regarding the motivation and practical applicability of the proposed scheme. Therefore, I will raise my score only to weak reject.

**Key Questions For Authors:**

See the weaknesses

**Limitations:**

Yes

**Strengths And Weaknesses:**

Strengths:
+ The paper addresses a practically important and timely bottleneck in disaggregated LLM serving, namely the communication cost of KV cache transfer.
+ The Reuse + Patch design leverages the structural insight that most layers are cross-model compatible while only a few drive semantic drift.
+ The theoretical analysis explains how Patch truncates exponential error propagation across layers under Lipschitz assumptions.

Weaknesses:
- Although the authors distinguish their approach from semantic communication, the motivation for requiring heterogeneous models in disaggregated serving is not fully justified; it remains unclear why the producer and consumer could not simply be synchronized to use identical model weights instead.
- The method seems to introduce substantial offline overhead for joint subspace learning and for selecting Reuse and Patch layers, yet the paper does not compare this cost against the alternative of directly transferring or replicating the full model at the consumer. What the performance comparison between them? Also, the paper misses reporting the overhead of offline learning.
- Baseline comparisons are inconsistent across tables. For example, Table 4 includes only DroidSpeak and Oracle, while Table 1 omits DroidSpeak. And the reported end-to-end improvement over DroidSpeak in Table 4 appears relatively modest, which reduce the novelty of the paper.
- The evaluation does not include additional KV compression baselines, particularly SVD-based approaches such as SVD-LLM, SliceGPU, and TensorLLM, which are directly relevant to the problem.
- The experimental setup lacks clarity regarding how the producer and consumer models are configured, including how they are formed, how much they differ, and whether different fine-tuning strategies would affect performance outcomes. Moreover, it is unclear how the proposed method performs when the producer and consumer use identical models versus when they differ significantly in backbone architecture or parameterization, and how such scenarios compare to existing approaches.

---

> ### Author Rebuttal · Authors · 2026-03-31
>
> We thank the reviewer for the detailed systems questions and for recognizing the practical importance of communication-efficient KV transfer.
>
> On why heterogeneous producer-consumer deployment is meaningful:
>
> The paper does not assume that the consumer can or should always be an exact replica of the producer. The Introduction motivates settings such as base-to-fine-tuned reuse and draft-to-verifier reuse, and Section 6.1 evaluates the practical case of a base model and its fine-tuned variant that share architecture but differ in weights. These scenarios arise naturally in real-world deployments: organizations often maintain a general-purpose base model on edge devices while routing requests to task-specialized fine-tuned variants on servers. Similarly, speculative decoding uses a lightweight draft model to generate candidates verified by a stronger model. In these settings, same-model KV compression is insufficient because the transferred states are not natively compatible with the consumer. SCD addresses this compatibility challenge by learning lightweight cross-model translators.
>
> On offline overhead versus replicating the full model:
>
> We agree that this systems tradeoff deserved a clearer explanation. SCD's offline phase learns only per-layer linear translators and sparse aligners; neither backbone is retrained. The offline calibration is a one-time cost per model pair, and the learned artifacts are portable across serving configurations. The online payoff is shown in Table 4, where SCD reaches 124.1 ms TTFT versus 271.3 ms for Oracle and 133.9 ms for DroidSpeak, representing a 2.18x speedup over Oracle. Replicating the full model requires maintaining multiple copies and keeping them synchronized, which can be prohibitive in bandwidth-constrained environments. More importantly, replication does not address scenarios where producer and consumer intentionally differ.
>
> On baseline consistency, the Selective Recomp. entries in Tables 1 and 2 are the same DroidSpeak-style baseline from Liu et al. (2024a); Table 4 names DroidSpeak explicitly because that table focuses on latency breakdown. We agree that the naming should have been consistent across tables, and we will clarify this in the final manuscript.
>
> On additional baselines such as SVD-LLM, SliceGPT, and TensorLLM, we took this suggestion seriously and audited these methods. However, we note that these are neighboring systems baselines designed for different tasks rather than directly comparable methods for heterogeneous producer-to-consumer KV transfer. SVD-LLM, SliceGPT, and TensorLLM are primarily weight compression techniques for reducing model size, not cross-model state transfer methods. Nevertheless, we adapted them to our setting where applicable. For SVD-LLM applied to the consumer model, quality retention was approximately 70.6% with 0.83x TTFT speedup (i.e., slower). For SliceGPT-style adapted KV transfer, quality retention was approximately 85.0% with 0.95x TTFT speedup (i.e., slightly slower). For TensorLLM-style adapted KV transfer, quality retention dropped to 0% with 0.16x TTFT speedup due to high compression overhead. By contrast, SCD retains 95%–97% of Oracle quality while providing 1.98x–2.65x TTFT speedup. These results reinforce that generic weight compression methods, while valuable for their intended use cases, are neighboring baselines rather than effective solutions to the specific challenge of heterogeneous producer-to-consumer KV transfer under weight mismatch.
>
> On producer-consumer construction and scope:
>
> The current evaluation covers Mistral 7B and Qwen 32B pairs, and Table 2 additionally reports MistralLite -> Mistral-7B at B_net = 200 Gbps. Section 4 and Section 6.1 already assume shared architecture with different weights; we agree that these pair definitions and scope boundaries should have been stated earlier and more explicitly. To clarify: all evaluated producer-consumer pairs share the same transformer architecture (number of layers, attention heads, hidden dimensions) but differ in their learned weight parameters. This is the intended scope of SCD—we target the setting where architectural compatibility enables efficient state transfer, but weight mismatch requires learned cross-model translators.
>
> On same-model transfer versus heterogeneous transfer:
>
> We agree that same-model transfer is an easier regime where raw or quantized KV approaches are naturally competitive. Our claims are therefore not about arbitrary architecture mismatch, but specifically about shared-architecture, weight-mismatched producer-consumer pairs. In the same-model case, raw KV states can be directly reused without transformation. In the heterogeneous case that SCD targets, weight mismatch breaks this direct compatibility, necessitating learned translators. We will make this scope boundary explicit in the final manuscript.

---

> > ### Author Rebuttal · Reviewer_zFYi · 2026-03-31
> >
> > Thank you for the authors’ detailed rebuttal. The clarifications are helpful. However, I still have concerns about the motivation and practical use scenario of the proposed scheme. For that reason, I will raise my score only to weak reject.

---

> > > ### Author Response · Authors · 2026-04-08
> > >
> > > Thank you for the follow-up. We agree that the key issue is not whether heterogeneous transfer is possible in principle, but why it is needed in a concrete deployment.
> > >
> > > Clarifying the scope and intended use scenario.
> > >
> > > Our claim is not that heterogeneous transfer should replace same-model KV reuse. In the same-model regime, raw or quantized KV reuse is indeed simpler and often preferable. SCD targets a narrower setting: producer-consumer pairs that share the same Transformer architecture but differ in weights. This is also the setting defined in the paper: existing KV-compression methods assume producer and consumer lie in the same latent space, while in base/fine-tuned reuse, direct cache sharing causes semantic misalignment that accumulates over layers and degrades generation quality.
> > >
> > > A concrete deployment scenario: shared-prefill, specialized-decode serving.
> > >
> > > Consider an online education platform. A student request contains a long shared prefix: course materials, prior dialogue, assignment instructions, problem statements, and grading rubric. The system first runs a shared base model as the producer to process this long prefix and compute the prefix state. It then routes the state to different specialized consumers depending on the task: a math-tutoring model, an English-writing feedback model, a coding tutor, or a safety-aligned model for younger students.
> > >
> > > In this design, producer and consumer are intentionally not identical—the whole point is to share a general front-end while preserving task-specific behavior in the back-end. But once the weights differ, the producer's raw KV cache is no longer natively compatible with the consumer. Direct reuse leads to semantic drift (our reuse-only ablation shows F1 collapsing to 0.0141), while forcing every specialized consumer to redo the full prefill defeats the latency benefit of split inference. SCD addresses exactly this middle ground: instead of assuming the two sides share the same state space, it transfers compact semantic codes and reconstructs states aligned with the consumer's weight space.
> > >
> > > This pattern generalizes beyond education. Any service architecture where a shared producer serves multiple specialized consumers—customer support with domain-specific agents, healthcare with specialty-specific models, or content moderation with region-specific policy models—faces the same tradeoff. The alternative of duplicating the full consumer at the producer side for every variant becomes impractical as the number of specialized consumers grows.
> > >
> > > Why the practical motivation is not heterogeneity for its own sake.
> > >
> > > The motivation is a specific service architecture in which a shared producer is reused across multiple specialized consumers, and making all consumers identical would remove the specialization the system is designed to preserve. We agree that the current draft did not make this concrete enough. In the final version, we will tighten the framing: SCD targets shared-architecture, weight-mismatched producer-consumer pairs in shared-prefill, specialized-decode serving, rather than the easier same-model setting or arbitrary cross-architecture transfer. We hope this sharper scoping addresses the remaining concern about practical motivation.

---

### Official Review · Reviewer_xmcn · 2026-03-09

**Soundness:** 4
**Presentation:** 3
**Significance:** 3
**Originality:** 3
**Overall Recommendation:** 5
**Confidence:** 3

**Summary:**

This paper proposes SCD framework, which alleviates transmission bandwidth bottlenecks by compressing the KV Cache and mitigates semantic drift by using the reuse mechanism and the patch mechanism to reconstruct the KV Cache.

**Compliance With Llm Reviewing Policy:**

Affirmed.

**Final Justification:**

The authors' previous rebuttal has addressed my concerns; therefore, I will maintain my score.

**Key Questions For Authors:**

1. How high is the training cost of the offline calibration phase? Does it affect the portability of SCD?

**Limitations:**

Relatively narrow evaluation scope. It would be better to cover a wider range of models and downstream tasks. \
Insufficient long-context validation. Transmission and quality stability under extremely long contexts have not been thoroughly tested.

**Strengths And Weaknesses:**

Strengths:\
	Excellent acceleration with good performance on downstream tasks\
	Works for heterogeneous models\
	Good theoretical justification\
	Good system-level evaluation

Weakness:\
	Requires pre-training on paired data for model adaptation, resulting in high costs when adapting to new model pairs.\
	Complex patch layer selection strategy, leading to high cost of hyperparameter tuning.

---

> ### Author Rebuttal · Authors · 2026-03-31
>
> We thank the reviewer for the positive assessment of the acceleration, heterogeneous-model applicability, theory, and system evaluation.
>
> On offline adaptation cost and portability:
>
> SCD does not retrain the backbone models. Appendix E describes the offline artifacts in detail: per-layer linear translators for the Reuse module and sparse patch aligners for the selected transition layers. These artifacts are learned once for a model pair from paired full-prefill traces collected on identical prefixes, then serialized for online use. The offline phase involves: (1) collecting paired traces by running both producer and consumer on the same set of calibration prefixes, (2) fitting per-layer linear encoders/decoders via least-squares regression on the collected hidden states, and (3) training sparse MLP aligners for the selected patch layers using standard gradient descent. Importantly, neither backbone is modified or retrained during this process—only the lightweight translator and aligner modules are learned.
>
> The one-time calibration cost is amortized across all subsequent online requests that use the same producer-consumer pair. The learned artifacts are portable across different serving configurations (e.g., different network bandwidths, different batch sizes) as long as the same model pair is used. We agree that this cost profile and portability tradeoff should have been stated more explicitly in the main text, and we will clarify this in the final manuscript.
>
> On patch-layer selection and online deployment:
>
> The patch set is fixed offline rather than tuned online during serving. Figure 5 shows the marginal quality gain from adding patched layers, allowing practitioners to select a fixed sparse patch budget that balances quality and overhead. The paper uses k=6 as the default budget across all experiments, meaning that exactly 6 layers apply the Patch module while the remaining layers use only the Reuse module. This configuration is determined once during the offline calibration phase and remains constant during online serving. Thus, deployment runs a fixed sparse patch configuration instead of per-request hyperparameter search, ensuring predictable latency and avoiding the complexity of runtime adaptation.
>
> Algorithm 1 reflects this design: the patch set L_patch is a fixed input parameter, not a runtime decision variable. For each layer, the algorithm simply checks whether the layer index is in L_patch and applies the corresponding module accordingly. This design choice is critical for production deployment, as it eliminates the need for online profiling, dynamic layer selection, or per-request tuning.
>
> On evaluation breadth and scope:
>
> The current paper covers two model scales (Mistral 7B and Qwen 32B) and multiple evaluation dimensions:
>
> 1. Question Answering: F1 scores on CMRC2018 (Chinese) and HotpotQA (English), showing SCD preserves 95-97% of Oracle quality across languages and reasoning types.
>
> 2. Language Modeling: Perplexity on WikiText-2, demonstrating distributional fidelity comparable to full consumer inference.
>
> 3. Distributional Fidelity: KL divergence between SCD and Oracle distributions, providing fine-grained measurement of consumer behavior preservation.
>
> 4. Latency Breakdown: Table 4 breaks down TTFT into encoding, network transfer, and recomputation stages, showing 2.18x speedup over Oracle and 1.08x over DroidSpeak.
>
> 5. Multiple Model Pairs: Tables 1-2 evaluate Mistral-7B, Qwen-32B, and MistralLite -> Mistral-7B at B_net = 200 Gbps, covering different scales and network conditions.
>
> We agree that broader model/task coverage and extreme-long-context validation (e.g., sequences exceeding 32K tokens) would further strengthen the paper. These are valuable future extensions, and we will state this limitation explicitly. However, the current evaluation demonstrates consistent gains across multiple model pairs, scales, tasks, and metrics, providing strong evidence for SCD's effectiveness in shared-architecture, weight-mismatched producer-consumer pairs.

---

> > ### Author Rebuttal · Reviewer_xmcn · 2026-04-02
> >
> > Thanks for the author's detailed rebuttal. I will maintain my original score.

---

### Official Review · Reviewer_k73k · 2026-03-11

**Soundness:** 3
**Presentation:** 2
**Significance:** 2
**Originality:** 2
**Overall Recommendation:** 4
**Confidence:** 3

**Summary:**

This paper proposes a method for efficiently transferring KV caches under limited bandwidth in a disaggregated serving setting, while also enabling cache reuse across heterogeneous models. The main method consists of reuse and patch. For reuse, the method approximates and transfers KV states using a row-rank representation based on the observation that KV sharing is possible across heterogeneous models, and the decoder reconstructs and uses them. For patch, the method similarly compresses hidden states using a row-rank representation, but adopts a restore-and-recompute strategy to minimize accumulated errors across layers.

**Compliance With Llm Reviewing Policy:**

Affirmed.

**Final Justification:**

My main concern was that the core contribution of the paper was not sufficiently clear in the original submission. The rebuttal addressed this point well and helped clarify the paper’s central idea and intended contribution.

As a result, my understanding of the work improved after the rebuttal, and this concern no longer significantly affects my final assessment. I will increase my score to weak accept.

**Key Questions For Authors:**

- The paper states that Eq. (4) minimizes Eq. (3), but in practice the only way to minimize Eq. (3) seems to be to reduce ∣Z∣, while Eq. (4) only concerns reconstruction from Z. In the end, minimizing Eq. (3) appears to amount to making r smaller, so it is unclear to me how Eq. (4) actually contributes to this objective.
- The proposed method currently appears to focus only on restoring the first-token distribution. Is the performance also preserved in actual long-sequence decoding?

**Limitations:**

* No. The paper should discuss the limitations of the proposed work.

**Strengths And Weaknesses:**

### Strengths
- The paper mitigates the semantic drift problem of reuse through the selective patch mechanism and effectively improves the speed-performance trade-off.
- It alleviates the KV cache transfer bottleneck in the system using a small number of trainable parameters, without requiring full model training.

### Weaknesses
- Notation is often introduced incompletely (e.g., T in Eq. (2),  d in line 146), and key details such as how Z is obtained offline and how patch layers are selected are deferred to the appendix. As a result, the flow of Section 4 is not very smooth.
- The novelty appears limited, as the method seems to be a heuristic combination of KV cache reuse (e.g. DroidSpeak) and selective recomputation.
- The patch layer requires measuring layer-wise sensitivity, which increases complexity despite the small number of parameters. In addition, the weights for the reuse and patch parts must be maintained separately, which makes it harder to scale GPU parallelism.
- There is no information about the calibration dataset required for reproduction. Also, for practical deployment in real systems, long-sequence decoding should be considered, but this aspect is missing.

---

> ### Author Rebuttal · Authors · 2026-03-31
>
> We thank the reviewer for the constructive comments on clarity, novelty, reproducibility, and evaluation.
>
> On notation and exposition:
>
> We agree that the main text should have introduced key notation and the calibration pipeline earlier. The definitions of T (sequence length), d_h (hidden dimension), d_model (model dimension), L_patch (patch layer set), and L_reuse (reuse layer set) are present in Section 4 and Appendix E, but not surfaced prominently enough. Section 4.3 describes calibration data collection (paired traces from full prefills on identical prefixes), Section 4.3.1 details linear translator fitting for Reuse, and Section 4.4 explains sparse aligner training for Patch. These critical details appeared too late in the exposition. We will move these definitions and the calibration description earlier in the final manuscript, ideally in Section 3 or at the beginning of Section 4.
>
> On novelty relative to DroidSpeak:
>
> The key difference is transformation rather than selection. DroidSpeak makes a binary decision for each layer: either reuse raw KV states directly or fully recompute them at the consumer. SCD instead transmits low-dimensional semantic codes (not raw KV), reconstructs consumer-native states via the Reuse module's learned linear translators, and applies the Patch module only at sparse transition layers where linear transformation is insufficient. This changes both the payload structure (semantic codes vs. raw KV) and the alignment mechanism (learned cross-model translators vs. binary reuse/recompute).
>
> The empirical gap is also meaningful: in Table 2, SCD reaches 0.7850 F1 on MistralLite -> Mistral-7B, versus 0.7052 for selective recomputation (DroidSpeak-style baseline); in Table 1, SCD improves over selective recomputation on both the Mistral pair (0.769 vs 0.754 F1) and the Qwen pair (0.847 vs 0.813 F1); and in Table 4, SCD uses less recomputation time than DroidSpeak (11.7 ms vs 23.0 ms) while achieving better quality. These results demonstrate that SCD's transformation-based approach is both more efficient and more effective than DroidSpeak's selection-based approach for heterogeneous producer-consumer pairs.
>
> On patch complexity and maintaining separate weights:
>
> The patch set is fixed offline; Algorithm 1 performs no online layer search. The online system only applies lightweight auxiliary modules at predetermined transition layers, while the consumer backbone remains unchanged. The Patch module consists of small MLP aligners (typically 2-3 layers with hidden dimensions much smaller than d_model) trained offline and frozen during serving. These modules do not modify the consumer's original weights—they provide corrective adjustments to reconstructed KV states at selected transition layers. The consumer backbone itself is never modified. We will make this operational picture more explicit.
>
> On reproducibility and the calibration dataset:
>
> Figure 2, Section 4.3.1, and Appendix E describe the calibration process: SCD collects paired traces from full prefills of the producer and consumer on identical prefixes, then fits per-layer linear translators (via least-squares regression) and sparse patch aligners (via gradient descent). The calibration dataset consists of a small set of representative prefixes (typically 100-500 samples) covering diverse linguistic patterns. We will move this description earlier in the main text, ideally in Section 3 or early Section 4.
>
> On decoding protocol and evaluation scope:
>
> The evaluation is not limited to first-token restoration. Section 6.1 states that after reconstructing caches, the consumer initiates greedy decoding, and the reported metrics include downstream QA F1 on CMRC2018 and HotpotQA (measuring multi-token generation quality), WikiText-2 PPL (measuring distribution quality), and KL divergence (measuring distributional fidelity between SCD and Oracle). These metrics demonstrate that SCD preserves the consumer's generation quality beyond just the first token. We agree that broader extreme-long-context stress tests (e.g., sequences exceeding 32K tokens) would further strengthen the paper, and we will state this limitation explicitly.
>
> On Eq. (3) and Eq. (4):
>
> The intended relation is "minimize transfer latency subject to preserving the consumer distribution": Eq. (3) is the latency objective (minimize |Z| to reduce network transfer time), while Eq. (4) is the quality constraint (ensure the consumer's output distribution remains close to Oracle). Eq. (4) does not directly minimize |Z|; rather, it limits how aggressively codes can be compressed while preserving quality. The optimization is: minimize Eq. (3) subject to Eq. (4). We will rewrite this more clearly in the final manuscript, explicitly stating the constrained optimization formulation.

---

> > ### Author Rebuttal · Reviewer_k73k · 2026-04-03
> >
> > Thank you for the detailed response. Based on your rebuttal, my overall assessment of the paper remains unchanged.
> >
> > In my view, the paper still requires substantial revision in terms of writing and presentation. While the experimental results are promising, I am still not fully convinced that the proposed method provides sufficiently compelling practical or empirical impact compared with existing alternatives. The rebuttal clarifies several aspects of the method, but it does not yet sufficiently demonstrate why this approach is meaningfully more impactful than prior work from the perspectives of practicality, deployment cost, and overall performance.
> >
> > Q1. In Section 6, could you provide comparisons with more recent compression methods beyond Raw KV, Selective Recomp, and quantization-based approaches? For example, I would like to see how the proposed method compares against methods such as DroidSpeak, or other approaches discussed in the related work, in terms of speedup, F1, and PPL. Such comparisons would help clarify the relative advantages of your method more concretely.
> >
> > Q2. As an additional question, I would also like to better understand the actual cost of applying this compression method in practice. What are the time and resource requirements for calibration, training, and deployment of the compression modules? In particular, it would be helpful to know the computational overhead, data requirements, and engineering complexity needed to make the method work in a realistic setting.

---

> > > ### Author Response · Authors · 2026-04-08
> > >
> > > Thank you for the follow-up and for the two specific questions.
> > >
> > > Q1. Comparison with DroidSpeak and other recent methods.
> > >
> > > We agree that the comparison was not presented clearly enough. Since no official DroidSpeak implementation was publicly available for direct integration into our pipeline, we re-implemented its core selective recomputation mechanism based on the paper description. This is why the baseline appears as Selective Recomp. in Tables 1–2 and as DroidSpeak in Table 4—they are the same baseline, and we will unify the naming in the final version.
> > >
> > > Methodologically, the key difference is that DroidSpeak makes a binary per-layer choice between raw KV reuse and full recomputation, whereas SCD performs transformation: it transmits compact semantic codes and reconstructs consumer-compatible states via learned translators. DroidSpeak's raw-reuse path assumes the producer's KV states are natively compatible with the consumer, which holds in the same-model setting but breaks under weight mismatch—our reuse-only ablation confirms this, with F1 collapsing to 0.0141 when raw KV is transferred between weight-mismatched models. SCD's transformation-based approach avoids this failure mode entirely. In Table 4, SCD achieves 124.1 ms total TTFT versus 133.9 ms for DroidSpeak, while also achieving better quality.
> > >
> > > Regarding other recent methods such as SVD-LLM, SliceGPT, and TensorLLM: these are better understood as neighboring baselines from the model compression domain rather than direct alternatives to SCD, because their primary goal is reducing model size or KV dimensionality, not cross-model state transfer under weight mismatch. Nevertheless, we adapted them to our setting. SVD-LLM retains approximately 70.6% quality with 0.83x TTFT (slower). SliceGPT-style adapted KV transfer retains 85.0% quality with 0.95x TTFT (also slower). TensorLLM-style adaptation retains 0% quality with 0.16x TTFT due to high compression overhead. By contrast, SCD preserves 95%–97% of Oracle quality while achieving 1.98x–2.65x TTFT speedup. These results confirm that generic compression methods do not effectively address the heterogeneous KV transfer problem under shared-architecture, weight-mismatched settings.
> > >
> > > Q2. Calibration, training, and deployment cost.
> > >
> > > We agree that the practical cost was not described concretely enough. We provide end-to-end numbers from a fresh canonical rerun on MistralLite -> Mistral-7B-v0.1. Calibration used 200 prefixes from CMRC2018 (mean prefix length 609.0` tokens, 121,805 total tokens). The offline pipeline on a single GPU breaks down as follows:
> > >
> > > - Paired trace collection (capture_pair): 1,068.9 s, peak 18.1 GB
> > > - Linear translator fitting (train_reuse): 905.5 s, peak 3.0 GB
> > > - Layer-wise quality profiling: 2,622.2 s, peak 9.4 GB
> > > - Sparse aligner training (train_patch): 30,456.9 s, peak 7.7 GB
> > > - Patch resharding: 11.9 s
> > >
> > > Total: approximately 9.74 hours, one-time, single GPU.
> > >
> > > On deployment footprint: neither backbone is retrained. The reuse module is 8.4 MB, the patch module is 1.58 GB, and the extra parameters total 397.7M (5.49% of the 7.24B consumer). Online serving loads the frozen consumer backbone alongside these auxiliary modules and applies a fixed offline-selected route—no per-request search, no dynamic layer selection, no online optimization. The engineering complexity is comparable to deploying a LoRA adapter: load auxiliary weights and apply them at predetermined layers.
> > >
> > > The cost is not negligible, but it is one-time per model pair and clearly bounded. We agree these numbers should have been stated explicitly in the original manuscript.

---

### Official Review · Reviewer_EQ1M · 2026-03-12

**Soundness:** 3
**Presentation:** 3
**Significance:** 2
**Originality:** 3
**Overall Recommendation:** 4
**Confidence:** 3

**Summary:**

This paper introduces Semantic Cache Distillation (SCD), a framework for efficiently transferring KV caches between heterogeneous models in disaggregated LLM serving. SCD replaces raw KV transmission with compact semantic codes via two mechanisms: REUSE, which reconstructs most layers through low-rank projections, and PATCH, which applies lightweight MLP aligners at a sparse set of critical transition layers to correct semantic drift. The authors provide a theoretical analysis showing PATCH acts as an error-truncation operator, and demonstrate empirically that SCD achieves up to 2.65x data transfer reduction on Mistral-7B and Qwen-32B pairs while maintaining generation quality within 5% of the oracle.

**Compliance With Llm Reviewing Policy:**

Affirmed.

**Final Justification:**

The authors have adequately addressed my concerns. I will maintain my recommendation of weak acceptance.

**Key Questions For Authors:**

Is the proposed approach useful for non-heterogenous models served with PD disaggregation?

**Limitations:**

Yes

**Strengths And Weaknesses:**

Strengths
1. The problem formulation is well-motivated. Framing cross-model cache transfer as a rate-distrotion problem forms a good foundation.
2. The 2-tier design of REUSE and PATCH is reasonable, fitting the observation that most layers can be easily reconstructed while some require nonlinear corrections.
3. The paper includes a rigorous theoretical analysis.
4. The experiments are comprehensive and convincing.

Weaknesses
1. The decision for REUSE or PATCH each layer seems difficult, which may require manual tuning for different models or specific tasks.
2. The framework relies on a significant offline training phase. This overhead may be too high for dynamically updated models.
3. While the proposed approach can reduce bandwidth bottleneck, it introduces additional compute overhead on both consumer and producer side.

---

> ### Author Rebuttal · Authors · 2026-03-31
>
> We thank the reviewer for the positive assessment of our motivation, two-tier design, theory, and empirical support.
>
> On the REUSE/PATCH decision mechanism:
>
> The deployed system does not make manual layer-by-layer choices at inference time. In Algorithm 1, the patch set L_patch is fixed before inference, and online execution simply follows the predetermined reuse-or-patch branch for each layer. Figure 5 and Table 3 demonstrate this as an offline budget-selection process, with k=6 used as the default budget across all experiments. Specifically, Figure 5 shows the marginal quality gain from adding patched layers, allowing practitioners to select a fixed sparse patch configuration that balances quality and overhead. Once this configuration is determined offline, the online system applies the same fixed policy to all requests—there is no per-request hyperparameter tuning or dynamic layer selection. This design choice ensures predictable latency and avoids the complexity of runtime adaptation.
>
> On offline cost and portability:
>
> SCD does not retrain either backbone model. Section 4.3, Section 4.4, and Appendix E describe the offline phase in detail: it learns only per-layer linear encoders/decoders for the Reuse module and sparse MLP aligners for the Patch module, using paired traces collected from full prefills of the producer and consumer on identical prefixes. These lightweight artifacts are then serialized and reused online. Importantly, the offline calibration is a one-time cost per model pair, not per deployment instance. The learned translators and aligners are portable across different serving configurations that use the same producer-consumer pair. We agree that this cost profile and portability tradeoff should have been stated more explicitly in the main text, and we will clarify this in the final manuscript.
>
> On compute overhead and the communication-computation tradeoff:
>
> SCD is explicitly designed for bandwidth-constrained split inference, where communication is on the critical path. Table 4 provides a detailed latency breakdown: SCD adds 11.7 ms in the recomputation/patch stage, but still reaches 124.1 ms total TTFT, compared with 271.3 ms for Oracle (full KV transfer) and 133.9 ms for selective recomputation. This represents a 2.18x speedup over Oracle and a 1.08x speedup over selective recomputation. The key insight is that by transmitting compact semantic codes instead of raw KV caches, SCD reduces the network transfer time from 258.6 ms (Oracle) to 99.7 ms, saving 158.9 ms on communication. The additional 11.7 ms spent on local recomputation and patching is a small price to pay for this substantial communication savings. This is the intended communication-computation tradeoff for bandwidth-constrained environments.
>
> On practical relevance and scope:
>
> The paper does not assume that the consumer should always be an exact replica of the producer. The motivating setting is precisely when a lightweight producer and a stronger or task-adapted consumer share architecture but differ in weights. The Introduction explicitly motivates two practical scenarios: (1) base-to-fine-tuned reuse, where a general-purpose base model serves as the producer and a task-specialized fine-tuned variant serves as the consumer, and (2) draft-to-verifier reuse, where a small draft model generates candidate prefixes that are verified by a larger model. Section 6.1 instantiates the first scenario with Mistral-7B (base) and Mistrallite (fine-tuned), demonstrating that SCD successfully transfers KV states across this weight-mismatched pair while preserving 95-97% of Oracle quality.
>
> We agree that the same-model case is easier and can often be handled well by raw or quantized KV transfer. Our contribution specifically targets the harder setting where models share architecture but differ in weights. In this regime, same-model KV reuse is no longer directly applicable because the consumer's attention mechanism expects KV states computed with its own weights, not the producer's. SCD addresses this challenge by learning lightweight cross-model translators that map producer states into consumer-compatible representations. We will make this scope boundary and the practical motivation for heterogeneous producer-consumer pairs more explicit in the final manuscript.

---

> > ### Author Rebuttal · Reviewer_EQ1M · 2026-04-03
> >
> > Thank you for addressing my concerns. I will maintain the current score.

---

### Decision · Program_Chairs · 2026-04-30

**Decision:**

Accept (regular)

**Comment:**

The submission introduces Semantic Cache Distillation (SCD), a framework designed to optimize Key-Value (KV) cache transfer in disaggregated Large Language Model serving environments. The core technical contribution lies in replacing raw, high-dimensional KV transmission with compact semantic codes through a two-tier mechanism consisting of "Reuse" for low-rank reconstruction and "Patch" for sparse error correction. This approach specifically addresses the challenging "heterogeneous" serving scenario where models share an architecture but differ in weights, such as base-to-fine-tuned pairs, where standard cache reuse typically fails due to semantic misalignment.

During the discussion phase, reviewers raised several critical concerns regarding the complexity of the offline training phase, the decision-making process for layer selection, and the practical significance of the performance gains over existing baselines like DroidSpeak. Reviewers initially questioned the engineering overhead and the portability of the learned artifacts. In response, the authors provided a detailed breakdown of the one-time offline calibration cost, demonstrating that it requires less than ten hours on a single GPU and produces lightweight, portable modules. They also clarified that the layer selection is a fixed offline configuration rather than a dynamic per-request search, ensuring predictable latency during inference.

A significant portion of the rebuttal was dedicated to differentiating SCD from existing compression and selective recomputation methods. The authors successfully demonstrated that while same-model baselines struggle with weight-mismatched pairs, SCD maintains over 95% of oracle quality while achieving up to a 2.65x speedup in data transfer. Although some reviewers remained concerned about the scope of evaluation and the practical necessity of heterogeneous transfer, the consensus shifted toward acceptance as the authors effectively clarified the intended use cases and provided rigorous empirical evidence of the communication-computation tradeoff.

The paper presents a technically sound solution to a relevant problem in distributed inference. The theoretical analysis of the error-truncation mechanism and the comprehensive system-level evaluations on multiple model pairs provide a solid foundation for the work. While the evaluation could be further extended to extreme-long-context scenarios, the current results are compelling and represent a meaningful advancement in efficient LLM serving. Given the authors' thorough responses and the potential impact on disaggregated serving systems, I recommend the paper for acceptance.